# Novel Exploration via Orthogonality

**Andreas Theophilou**
University of Bath
United Kingdom
ajt80@bath.ac.uk

**Özgür Şimşek**
University of Bath
United Kingdom
o.simsek@bath.ac.uk

## Abstract

Efficient exploration remains one of the most important open problems in reinforcement learning. Discovering novel states or transitions requires policies that efficiently direct the agent away from the regions of the state space that are already well explored. We introduce *Novel Exploration via Orthogonality (NEO)*, an approach that automatically uncovers not only which regions of the environment are novel but also how to reach them by leveraging Laplacian representations. NEO uses the eigenvectors of a modified graph Laplacian to induce gradient flows from states that are frequently visited (less novel) to states that are seldom visited (more novel). We show that NEO's modified Laplacian yields eigenvectors whose extreme values align with the most novel regions of the state space. We provide bounds for the eigenvalues of the modified Laplacian; and we show that the smoothest eigenvectors with real eigenvalues below certain thresholds provide guaranteed gradients to novel states for both undirected and directed graphs. In an empirical evaluation in online, incremental settings, NEO outperformed related state-of-the-art approaches, including eigen-options and cover options, in a large collection of undirected and directed environments with varying connectivity structures.

## 1  Introduction

Temporal abstraction has been widely studied as an approach to efficient exploration, which remains one of the key research challenges in reinforcement learning. State of the art methods include eigen-options [1, 2] and cover time options [3]. In symmetric settings, both of these methods can produce policies that push the agent toward the far reaches of the state space. However, such regions may already be heavily visited, and policies that push the agent naively to far away reaches of the environment are not always effective explorers.

An alternative approach is count-based exploration [4, 5, 6, 7]. This approach addresses novelty through pseudo-rewards, for example, by giving the agent a novelty bonus proportional to $\frac{1}{\sqrt{n}}$ for having visited a state-action pair $n$ times. Some algorithms that take this approach come with theoretical guarantees [4]. However, these algorithms do not provide explicit gradient guidance to the agent. Rather than providing explicit policies for exploration, they rely on primitive actions and standard reinforcement update rules, such as Q-value learning, to propagate their intrinsic rewards, which can introduce a lag in exploration until the agent's value estimates have sufficiently propogated.

A natural alternative is to construct policies that explicitly drive the agent toward the most novel states [8]. With access to an oracle solver for shortest paths in the transition graph, in symmetric settings, such policies can guarantee reaching a maximally novel state within a given horizon. However, this approach focuses exploration very narrowly on a single target, rather than diversifying across multiple novel regions.

We introduce *Novel Exploration via Orthogonality (NEO)*, a method that maintains the strengths of prior approaches whilst addressing their weaknesses. As with related spectral methods, NEO uses the

39th Conference on Neural Information Processing Systems (NeurIPS 2025).

Laplacian eigenvector's smooth and orthogonal property to encourage exploration towards distinct regions of the state space. Unlike prior methods, NEO introduces gradient guarantees and focuses the exploration towards regions that are novel.

Importantly, NEO stands out by providing Laplacian-based policies in directed settings. Previous Laplacian approaches assume an undirected transition graph, even though many real-world environments are inherently directed. Recent work has proposed using the polar decomposition of a transition matrix to recover real eigenvalues to obtain eigen-options in the directed setting [9]; however, this approach uses a symmetric, Hermitian matrix, which cannot provide directed gradient flows beyond the symmetric graph obtained in the construction. Moreover, eigen-options in directed settings by polar decomposition is used only as a baseline; it has not been shown in evaluations to perform desirably [9]. By careful construction, we obtain eigenvectors with gradient guarantees in strongly connected directed graphs for any eigenvector with an associated real eigenvalue below a given value. Conceptually, as we create a modified Laplacian and take its eigenvectors, we effectively convert novelty to energy where the most novel states dominate the contributions to the inner product $\langle u, v \rangle = \sum_i u_i v_i$ and are the highest energy states of the smoothest Laplacian eigenvectors with real eigenvalues below a novelty-based threshold.

We provide experimental results in a wide variety of environments, both directed and undirected, that illustrate the approach and its effectiveness. In online, incremental settings, NEO outperformed related state-of-the-art approaches, including eigen-options and cover options, in a large collection of undirected and directed domains with varying connectivity structures.[1]

## 2  Preliminaries

A finite Markov decision process (MDP) is a 5-tuple $\langle S, \mathcal{A}, \mathcal{T}, \mathcal{R}, \gamma \rangle$, where $S$ is a set of states, $\mathcal{A}$ a set of actions, $\mathcal{T}(s, a, s') \in [0, 1]$ the probability of transitioning to state $s'$ upon taking action $a$ in state $s$, $\mathcal{R}(s, a, s')$ the expected reward for that transition, and $\gamma \in [0, 1]$ the discount factor. At decision stage $t$, $t \geq 0$, the agent is in state $s_t \in S$, selects action $a_t \in \mathcal{A}$, and transitions to state $s_{t+1}$, receiving reward $r_{t+1}$. A *policy* $\pi(s, a)$ gives the probability of selecting action $a$ when in state $s$. The value function $V^\pi(s) = \sum_{a \in \mathcal{A}} \pi(s, a) \sum_{s' \in \mathcal{S}} \mathcal{T}(s, a, s') \Big[ \mathcal{R}(s, a, s') + \gamma V^\pi(s') \Big]$ is the expected discounted sum of future rewards when following policy $\pi$ from state $s$. An optimal policy is one that maximizes $V^\pi(s)$ for every state $s \in S$.

We refer to the actions of an MDP as *primitive actions* and represent *temporally-extended actions* using options [10, 11]. An *option* $o$ is a triple $\langle I_o, \pi_o, \beta_o \rangle$, where $I_o \subseteq S$ is the set of states in which the option can be initiated, $\pi_o$ is the policy followed during option execution, and $\beta_o : S \to [0, 1]$ is the *termination condition*, expressing the probability of termination in a given state.

Associated with an MDP $\langle S, \mathcal{A}, \mathcal{T}, \mathcal{R}, \gamma \rangle$, we define a directed graph $G = (V, E)$, where $V = S$ is the set of vertices and $E$ is the set of edges, with $(s, s') \in E$ iff $\exists a \in \mathcal{A}$ such that $\mathcal{T}(s, a, s') > 0$. Given this *transition graph*, we define *adjacency matrix* $A$, where $A_{ij} = 1$ if $(i, j) \in E$ and $0$ otherwise, and *random walk transition matrix* $P$, where $P_{ij} = \frac{A_{ij}}{\deg^+(i)}$, with $deg^+$ denoting the out-degree of a node, $deg^+(i) = \sum_j A_{ij}$.

The *Rayleigh quotient* of matrix $M \in \mathbb{C}^{n \times n}$ at vector $\mathbf{x} \in \mathbb{C}^n \setminus \{\mathbf{0}\}$ is the scalar $R(M; \mathbf{x}) = \frac{\mathbf{x}^* M \mathbf{x}}{\mathbf{x}^* \mathbf{x}}$, where $\mathbf{x}^*$ is the conjugate transpose of $\mathbf{x}$. We use $\lambda$ to denote the eigenvalues and $f$ the eigenvectors. If $M$ is Hermitian ($M = M^*$), then $R(M; \mathbf{x}) \in \mathbb{R}$ for all $\mathbf{x} \neq \mathbf{0}$, and $\lambda_{\min}(M) = \min_{\mathbf{x} \neq 0} R(M; \mathbf{x})$, $\lambda_{\max}(M) = \max_{\mathbf{x} \neq 0} R(M; \mathbf{x})$, with the extrema attained exactly at the corresponding eigenvectors $f$ of $M$. For a complex number, we use $\Re$ to refer to the real part and $\Im$ to refer to the imaginary part.

## 3  Source Laplacian

We start with defining *source Laplacian* $L_\zeta$ by adding non-negative diagonal weights to the *random-walk Laplacian* $L^{rw} = I - P$, thereby concentrating energy on selected nodes. Let $G = (V, E)$ be a directed, strongly connected graph of $n$ nodes, with row-stochastic transition matrix $P = $

---

[1]All code is available at `https://github.com/AndreasTheo/NEO`

$(P_{ij})$ : $P_{ij} \geq 0$, $\sum_{j=1}^{n} P_{ij} = 1$ $(\forall\, i)$. Let $I$ denote the identity matrix. Introduce nonnegative weights $\zeta = (\zeta_i)_{i=1}^{n}$ and define $n \times n$ matrix $\Gamma$, with $\Gamma_{ii} = \zeta_i$ and all other entries zero. Define $L_\zeta = I + \Gamma - P$. Adding $\Gamma$ increases the diagonal entries and, in eigenvectors of $L_\zeta$, pulls the eigenvector value at weighted nodes toward zero, raising the relative prominence of less weighted nodes where energy gets concentrated. A node $i$ with $\zeta_i = 0$ is a *source node*. A node $i$ with $\zeta_i > 0$ is a *sink node*. The Rayleigh quotient of source Laplacian $L_\zeta$ for any $x \in \mathbb{C}^n \setminus \{0\}$ is:

$$\mathcal{R}(x) = \frac{x^* L_\zeta x}{x^* x} = \frac{x^*(I + \Gamma)x - x^* P x}{x^* x} = \frac{\sum_{i=1}^{n}(1 + \zeta_i)|x_i|^2 - \sum_{i,j=1}^{n} \overline{x_i}\, P_{ij}\, x_j}{\sum_{i=1}^{n}|x_i|^2},$$

where $\overline{x_i}$ is the complex conjugate of $x_i$. If the graph is symmetric, the Rayleigh quotient can be expressed as:

$$\mathcal{R}(x) = \frac{x^\top L_\zeta\, x}{x^\top x} = \frac{\frac{1}{2}\sum_{i,j} P_{ij}(x_i - x_j)^2 \;+\; \sum_i \zeta_i\, x_i^2}{\sum_i x_i^2}$$

This is the standard Rayleigh quotient for a symmetric matrix $L^{rw} = I - P$ with the additional term $\sum_i \zeta_i\, x_i^2$, which is the added squared smoothness error contributions $\zeta_i(x_i - 0)^2$ between the node values $x_i$ and the zeros, weighted by $\zeta_i$.

Below, we present three theorems. In Theorem 3.1, we prove a lower bound for the eigenvalues of the source Laplacian and an upper bound for its smoothest eigenvalue. In Theorem 3.2, we show that, for certain eigenvalues, the maximum of the associated eigenvector is attained at a source node. In Theorem 3.3, for the source Laplacian with $\Gamma$ weighted by visitation values, we prove that, certain eigenvectors (those with associated eigenvalues lower than the maximum visitation value) give us guaranteed gradients (and thus paths) from nodes with higher visitation values to nodes with lower visitation values.

**Theorem 3.1** (Bounds for the eigenvalues $\lambda$ of the source Laplacian $L_\zeta = (I + \Gamma) - P$)**.** Suppose $k$ nodes are sources ($\zeta_i = 0$) and $n - k$ are sinks, each with uniform weight $\mu > 0$ (that is, $\zeta_i = \mu$ for all sinks). Then, $0 \leq R(\mathbf{1}) < \mu$, and the upper bound for the smoothest eigenvalue is $\mu$. If all sources are replaced with weight $\alpha$, where $\mu > \alpha \geq 0$, then $\alpha < R(\mathbf{1}) < \mu$, and the lower bound for the smoothest eigenvalue is $\alpha$ . If all $\zeta_i = \mu$, then $R(\mathbf{1}) = \mu$, and the smoothest eigenvalue is $\mu$.

*Proof.* $R(\mathbf{1}) = \frac{1}{n}\sum_{i=1}^{n} \zeta_i$. In the *standard case*, with $k$ sources and $n-k$ sinks, $k$ entries are zero and $n-k$ entries equal $\mu$, and $\sum_i \zeta_i = (n-k)\mu$. Consequently, $R(\mathbf{1}) = (n-k)\mu/n = (1-k/n)\mu$, which implies $0 \leq R(\mathbf{1}) < \mu$. In the $\alpha$*-weighted variant*, $\sum_i \zeta_i = k\alpha + (n - k)\mu$, so $R(\mathbf{1}) = \frac{k\alpha + (n-k)\mu}{n}$, a combination of $\alpha$ and $\mu$, hence $\alpha < R(\mathbf{1}) < \mu$. In the *uniform case*, $\sum_i \zeta_i = n\mu$, giving $R(\mathbf{1}) = \mu$. $\qquad\square$

**Theorem 3.2** (Maximality (in magnitude) of the eigenvector at sources)**.** Given source Laplacian $L_\zeta = (I + \Gamma) - P$, assume $L_\zeta f = \lambda f$ with real eigenvalue $\lambda < \min_{i:\zeta_i > 0} \zeta_i$, that is, the eigenvalue is less than the minimum non-zero weight in $\zeta$. Then the element of eigenvector $f$ with the highest absolute value, $|f_i|$, must be a source node ($\zeta_i = 0$).

*Proof.* From $L_\zeta f = \lambda f$ and $L_\zeta = I + \Gamma - P$, we have $(I + \Gamma - P)f = \lambda f$. Then for the $i^{\text{th}}$ row (corresponding to node $i$), after rearranging, we get $(1 + \zeta_i - \lambda)f_i = \sum_j P_{ij} f_j$, which yields:

$$f_i = \frac{1}{1 + \zeta_i - \lambda}\sum_j P_{ij} f_j \tag{1}$$

Taking absolute values at both sides of the equation, we obtain $|f_i| = \frac{1}{|1+\zeta_i-\lambda|}\left|\sum_j P_{ij} f_j\right|$. If the $i^{th}$ node is a sink node, then $\zeta_i > \lambda$, and $|1 + \zeta_i - \lambda| > 1$, and so

$$|f_i| \leq \frac{1}{1 + \zeta_i - \lambda}\sum_j P_{ij}|f_j| < \max_j |f_j|,$$

implying that, for a sink node $i$, $|f_i|$ cannot be a maximum entry of $f$ in magnitude. Therefore, maximality must be obtained at a source node. $\qquad\square$

**Theorem 3.3** (Novelty path via visit counts)**.** Let $G$ be a strongly connected graph, and $N(i)$ a visitation value of node $i$. Define $L_\zeta = (I + \Gamma) - P$ with $\Gamma_{ii} = N(i) \in \mathbb{R}_{\geq 0}$. For any eigenvalue $\lambda$ and associated eigenvector $f$ (right eigenvector $f$ if $L_\zeta$ is asymmetric) where $N(i) > \Re(\lambda)$ , there exists a directed path $v_0 \to v_1 \to \cdots \to v_k$ such that $|f_{v_0}| < |f_{v_1}| < \cdots < |f_{v_k}|$ and $N(v_k) < \Re(\lambda)$.

*Proof.* Following the same reasoning for the $i^{th}$ entry of $L_\zeta f = \lambda f$ as in Equation 1, we have $(1 + N(i) - \lambda)f_i = \sum_j P_{ij}f_j$. Therefore,

$$f_i = \frac{1}{1 + N(i) - \lambda} \sum_j P_{ij}f_j.$$

If $N(i) > \Re(\lambda)$, then

$$
\begin{aligned}
\left|1 + N(i) - \lambda\right| &= \left|(1 + N(i) - \Re(\lambda)) - \sqrt{-1}\,\Im(\lambda)\right| \\
&= \sqrt{(1 + N(i) - \Re(\lambda))^2 + \Im(\lambda)^2} \\
&\geq 1 + N(i) - \Re(\lambda) > 1.
\end{aligned}
$$

Therefore,

$$|f_i| = \frac{1}{\left|1 + N(i) - \lambda\right|} \left|\sum_j P_{ij}f_j\right| < \sum_j P_{ij}|f_j|,$$

implying $\sum_j P_{ij}|f_j| > |f_i|$. Thus some neighbor $j$ has $|f_j| > |f_i|$. Iterating yields a strictly increasing chain in $|f|$, terminating at $v_k$ with no neighbor with a larger magnitude, forcing $\left|1 + N(i) - \lambda\right| \leq 1$ and $N(v_k) < \Re(\lambda)$. $\qquad\square$

**Remark 3.1.** Because Theorem 3.3 uses only the magnitudes $|f_v|$, any complex phase of $f$ is irrelevant to the strict-inequality relations among $|f_{v_i}|$, $i = 0, ..., k$.

In the undirected setting, $L_\zeta$ is Hermitian and all its eigenvalues and eigenvectors are real. Consequently, one may simply take the eigenvectors corresponding to the $z$ smallest eigenvalues, knowing that each of these smoothest eigenvectors will define an orthogonal gradient field that carries mass from states with lower visitation values to states with higher visitation values, as long as the corresponding eigenvalue $\lambda$ is less than the maximum visitation value. When $G$ is directed, $L_\zeta$ becomes non-Hermitian, and complex eigenvalues (and eigenvectors) can appear. Nevertheless, any eigenvalue with $\Re(\lambda)$ below the maximum visitation value still induces a guaranteed gradient from nodes with higher visitation values to nodes with lower visitation values.

## 4   Novel Exploration via Orthogonality (NEO)

We now use the smoothest eigenvectors of the source Laplacian $L_\zeta$ to define multi-step exploration policies in the form of options. We call the proposed approach *Novel Exploration via Orthogonality*, or NEO, and outline it in pseudocode in Algorithm 1.

At each iteration of the algorithm, we start by following the agent's policy for $H$ steps, recording every transition so as to assemble (or update) an empirical graph with transition matrix $P$ and to accumulate raw visitation counts $N(s)$ for each state $s$ encountered during the roll out. Raw counts can vary wildly between instantiations; we therefore transform them via a scaling function $F: \mathbb{R} \to \mathbb{R}$, using two parameters: $\delta > 0, k > 0$. Specifically,

$$F\big(N(s)\big) = \delta \left(\frac{N(s)}{\max_{i \in S} N(i)}\right)^{1/k}, \tag{2}$$

where the denominator $\max_{i \in S} N(i)$ ensures that the maximum scaled count prior to being multiplied by $\delta$ is less than or equal to 1, the exponent $1/k$ compresses their dynamic range, and the multiplication with $\delta$ further shrinks or increases them. In Appendix B, we present results obtained by varying $\delta$ and

$k$. For all results presented in the main paper, we use $\delta = 0.5$ and $k = 64$. Using the scaled visitation counts $\zeta_i = F\big(N(i)\big)$, $i \in S$, we define an $n \times n$ matrix $\Gamma$, with $\Gamma_{ii} = \zeta_i$ and all other entries set to zero, and another $n \times n$ matrix $L_\zeta = I + \Gamma - P$, the source Laplacian.

We then compute the $Z$ smoothest eigenvectors of $L_\zeta$, those associated with its $Z$ smallest real eigenvalues. For every eigenvalue $\lambda < \delta$, the associated eigenvector $f$ defines a gradient that naturally points away from states that are heavily visited toward those that are less explored.

We instantiate one option per eigenvector. We use $f^o$ to denote the eigenvector that corresponds to option $o$. This option can be initiated in any state. The option policy moves the agent to the neighboring state with the largest magnitude of the corresponding eigenvector, $|f^o|$. The option terminates with probability 1 when the agent reaches a local peak of $|f^o|$ (that is, a state where the magnitude of the eigenvector exceeds or equals that of all its immediate neighbors), with probability zero elsewhere.

---

**Algorithm 1** NEO

---

1: **Input:** update horizon $H$, option count $Z$, scaling function $F$
2: $O = \{\}$       // set of options
3: $S = \{\}$       // set of states
4: **loop**
5:      Execute agent policy for $H$ steps
6:      Update state set $S$, graph $G$, transition matrix $P$, and visitation counts $N(.)$
7:      $L_\zeta = I + \Gamma - P$, where $\Gamma_{ii} = F(N(i))$
8:      Compute eigenpairs $\{(\lambda_o, f_o)\}_{i=1}^{Z}$ of $L_\zeta$, sorted by ascending $\lambda_o$
9:      **for** $i = 1$ to $Z$ **do**
10:        $I_o = S$                                        // initiation set
11:        $\beta_o(s) = 1$ iff $\forall s' : (s \to s'), |f_{s'}^o| \leq |f_s^o|$        // termination condition
12:        $\pi_o(s) =$ the action that leads to next state $s'$ that maximises $|f_{s'}^o|$    // option policy
13:        $o = \langle I_o, \pi_o, \beta_o \rangle$
14:        $O \leftarrow O \cup o$
15:      **end for**
16: **end loop**

---

In Figure 1, we illustrate the componentwise magnitude for eigenvectors of the source Laplacian in some directed and undirected domains. Plots (a) and (b) illustrate how the eigenvectors of the source Laplacian capture directed connectivity. In plot (a), we show a 20-node directed cycle, with one source ($\zeta = 0$; the node in the darkest shade of green) and all other nodes as equal sinks ($\zeta = 0.1$), yielding a smooth eigenvector that increases monotonically around the circle. In plot (b), we show a directed four-room grid with one-way doorways. Setting the top left corner as the source ($\zeta = 0$) and all other states as sinks ($\zeta = 0.1$) produces an eigenvector whose gradient reflects directed distance from sinks to the source.

Plots (c) and (d) show how we derive weights for $\Gamma$ from visitation counts in directed four rooms and in a maze environment, respectively. In directed four rooms, the agent performed a random walk of 100 steps, starting at the top left corner; this was repeated for 100 trials. In the maze, the agent performed a random walk of 10000 steps, starting at the bottom left corner; this was repeated for 500 trials. In both domains, visitation counts were scaled by their maximum, raised to the power of $\frac{1}{4}$ to compress their dynamic range, and multiplied by $\delta = 0.5$. In directed four-rooms, one state was never visited, whereas all states in the maze received some visitation (minimum $F(N(.)) \approx 0.475$).

Using the scaled visitation counts $F(N(.))$ to add to the diagonals of $\Gamma$, plots (e)–(h) display componentwise magnitude of the first four eigenvectors (scaled by $\sqrt[4]{.}$ for visualization) of the source Laplacian with novelty weighting for $\Gamma$ in directed four rooms; plots (i)–(l) show the same construction in the maze domain. We measured the cosine similarity between eigenvectors in directed four rooms (the eigenvectors displayed in plots (e)-(h)); the maximum value observed was 0.001. Orthogonality leads to distinct novel regions being high in magnitude across the eigenvector values. Notice that the smallest eigenvalue in plots (e) and (i) align with the theoretical lower and upper bounds set by the minimum diagonal value of $\Gamma$. We have a minimum $F(N(.))$ of 0 in plot (c) and 0.475 in plot (d) with a maximum $F(N(.))$ of 0.5 in both, resulting in eigenvalues of $\approx 0.136$ in plot (e) and $\approx 0.48$ in plot (i).

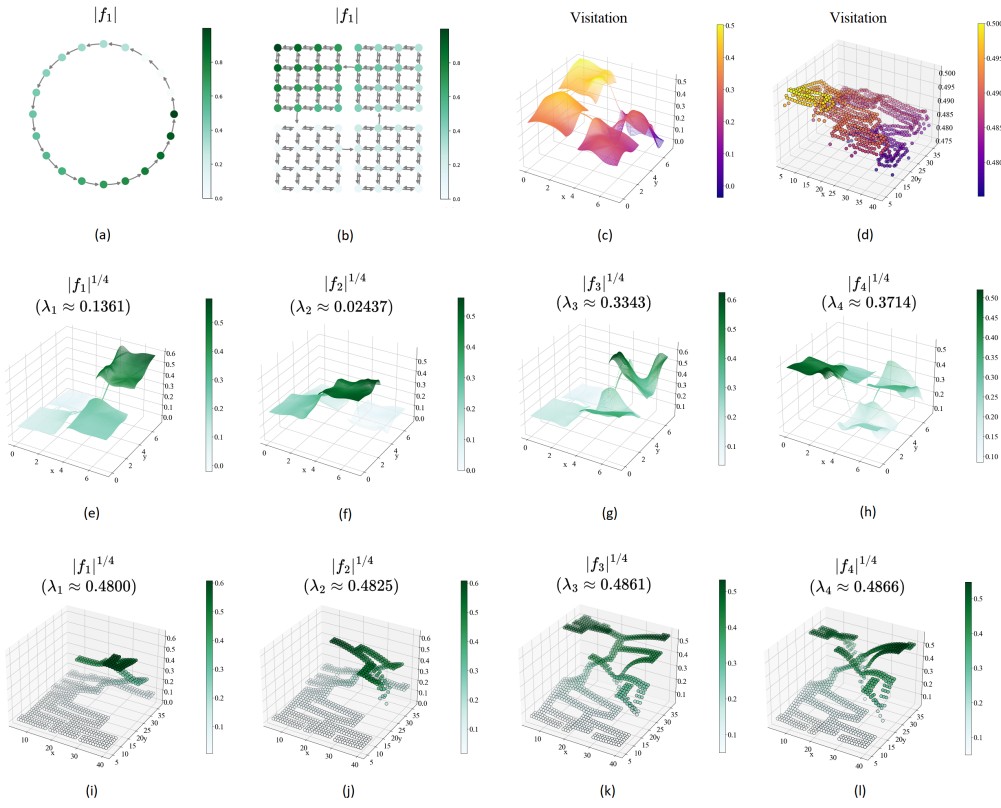

Figure 1: Smoothest eigenvector of $L_\zeta$ in (a) directed-cycle, (b) directed four rooms. Visitation-based sink weights in (c) directed four-rooms, (d) large maze. The first four eigenvectors of the source Laplacian $L_\zeta$ (scaled by $\sqrt[4]{\cdot}$ for visualization) in (e–h) directed four rooms, and (i–l) large maze. Please refer to Figure 2 for a visual depiction of these domains.

## 5 Related Work

Steinberger [12] demonstrated that the first eigenvector of an undirected graph Laplacian provides an approximate shortest-path solution via a single sink node, inspiring the spectral method proposed here. Earlier related works include eigen-options [1, 13, 2], which use eigenvectors of a normalized Laplacian for options policies, as well as proto value functions [14, 15], which use the same eigenvector construction for representation learning and efficient value estimation. Cover options [3] build on eigen-options, applying solely the fielder vector recursively. Diffusion options [9] use eigenvectors of the Laplacian to find diffusely separated option termination states; however, this method is not currently applicable in the online setting. The successor representation [16] has been shown to be closely related to the graph Laplacian [17]. Another set of closely related graph-based methods include sub-goal discovery via betweenness [18], graph cuts [19, 20], graph components [21], bridge centrality [22], stability centrality [23], graph clustering [24, 25], relative novelty [7], and modularity maximization [26]. Also related are methods for extending novelty counts to the deep reinforcement learning setting [6, 5, 27, 28, 29] and learning novelty based distances [30].

## 6 Empirical Evaluation

In Figure 2, we present a visual depiction of the domains used in our analysis. Those under 1000 states include `directed four rooms`, a gridworld with four rooms connected via one-way hallway states, `Rubik's cube end game`, the undirected subgraph of the $2 \times 2$ Rubik's cube containing

the states that are within 3 moves of the solved state, and the classic `tower of Hanoi` puzzle. At roughly 2500 states is the `office`, an undirected environment challenging to previous spectral-based exploration methods [26]. Larger still are `large maze` and `hex`, each with approximately 5500 states. In addition to these undirected versions, we created and experimented on directed versions of large maze and hex by adding a single directed edge between the two most distant states. Our largest domains are a directed `New York City (NYC)` street graph consisting of 10000 nodes and the `directed torus` domain. In directed torus, the red walls on the right side wrap in a one-way direction (from right to left) to the walls on the left side, and the blue walls at the top wrap in a one-way direction (top to bottom) to the walls at the bottom. This domain was constructed as a more challenging alternative to a standard torus, allowing us to assess how agents navigate when traversing any of a large number of directed edges that can preclude finding returning paths.

We use Q-learning with step size $\alpha = 0.4$, discount rate $\gamma = 0.99$, an $\epsilon$-greedy policy with $\epsilon = 0.1$, augmented with a growing library of temporally extended actions in the form of options. The initial Q values are set to $0$ for primitive actions and $-0.00001$ for options, ensuring that the agent policy initially favors primitives unless exploration explicitly invokes an option. When the $\epsilon$-greedy policy takes an exploration step, with probability $init_P$ the agent chooses a random option, otherwise a random primitive action. We fix $init_P$ at 0.1 for reward based evaluations in Figure 3; we evaluate the impact of the $init_P$ parameter in Figure 4. An option can be initiated in any state that was part of the graph used when the option was constructed (in other states, there would be no constructed option policy). Each transition triggers the usual one-step Q-learning update $Q(s, a) \leftarrow (1 - \alpha)Q(s, a) + \alpha\left[r + \gamma \max_{a'} Q(s', a') - Q(s, a)\right]$. Additionally, at the end of an episode, the entire episode's transitions are replayed in reverse order, with the same Q-learning update to accelerate reward propagation in our sparse-reward task. Option values are learned using SMDP

Q-learning backups: $Q(s, o) \leftarrow (1 - \alpha) Q(s, o) + \alpha\left[\sum_{j=0}^{\tau-1} \gamma^j r_{t+j+1} + \gamma^\tau \max_{a'} Q(s_{t+\tau}, a')\right],$

where $s$ is the state in which option $o$ was initiated, $\tau$ is option duration until termination, and $r_{t+j+1}$ are the rewards accrued while executing option $o$.

In all domains, there is a single goal state. Upon reaching the goal state, the agent receives a reward of $+100$ and the episode terminates. All other transitions give a reward of $0$. We use an adaptive episode horizon: 500 steps for domains under 1000 states, 1000 steps for domains with 501–3000 states, and otherwise the number of states in the domain. For each evaluation agent instance, a goal state is randomly chosen and the farthest state from the goal state is set to be the start state for all runs for the agent instance. For each method, we run 20 agent instances; we use a random seed (equal to run number) before selecting goal and start states, ensuring all start and goal states are the same across compared agents.

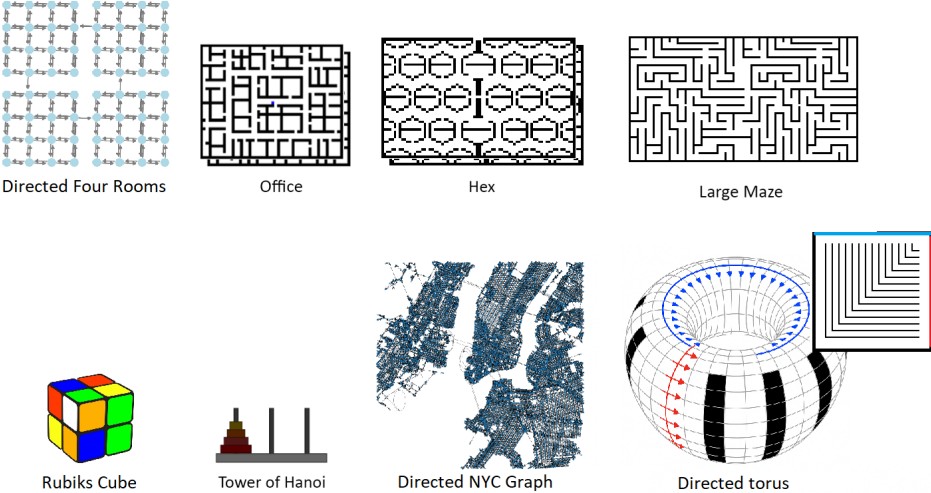

Figure 2: A visual depiction of the eight domains used in the empirical analysis.

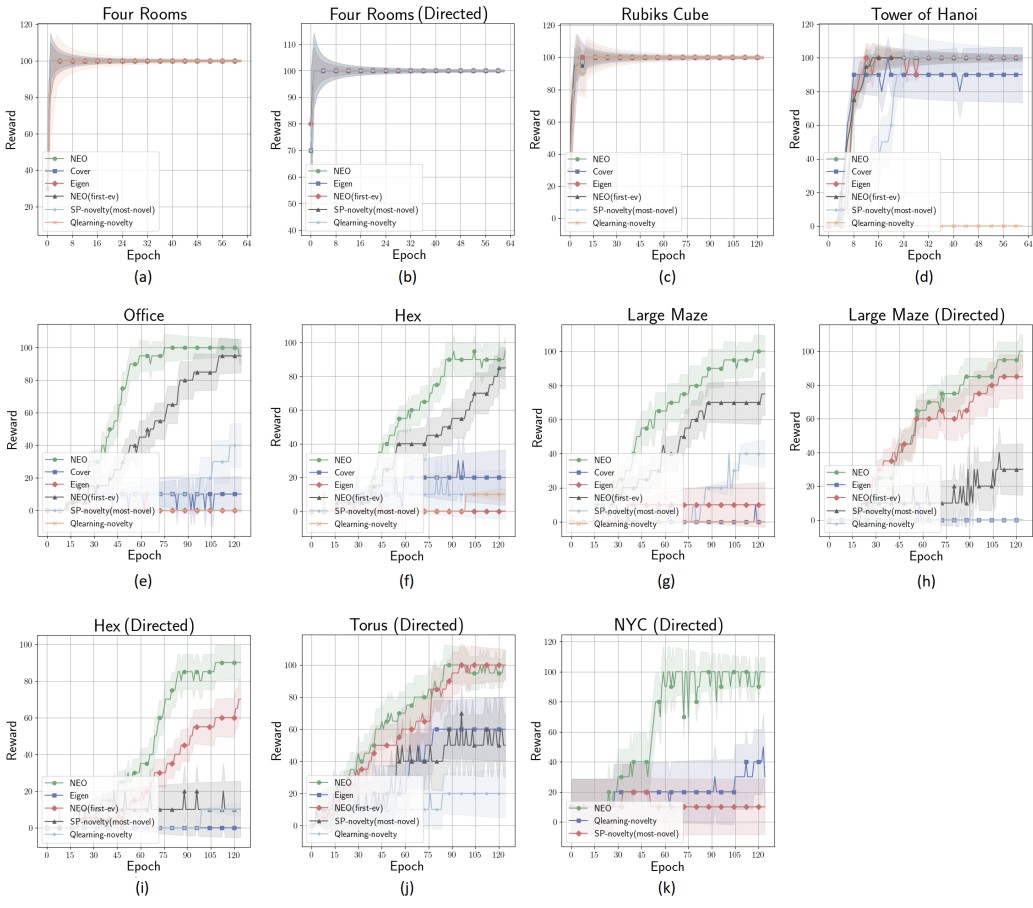

Figure 3: Comparative sparse reward empirical evaluation of learning agents with different option discovery methods across a set of domains.

Options are discovered online every $H$ decision stages, with $H$ set to five times the number of nodes in the domain's full transition graph. We generate four new options per update, capping the total stored options at 64 and discarding the oldest four options when capacity is exceeded and replacing them with the most recent four options. Each option policy is defined by ascending a representation until reaching a local maximum: for eigen-options, the representation is an eigenvector of the graph Laplacian; for cover-options, it is the Fiedler vector over graph nodes; for the shortest-path-novelty [8] options, it is the shortest path distance to the most novel node in the agent's working graph; and for NEO options, it is the magnitude of the eigenvectors described previously in Theorem 3.3. The eigenvectors can also be taken as intrinsic rewards for option policies instead of direct solutions. In Appendix C, we include the results for eigen-options and cover-options solved with policy iteration. In directed domains, the polar-decomposition variant of eigen-options are applied. We cannot use cover options in directed domains because they rely on the construction of Fiedler vectors.

Every 2000 steps, which we define as an *epoch*, we freeze exploration and run an independent evaluation episode where no exploration actions are taken ($\epsilon = 0$). We plot the evaluation results with a $2\sigma$ error width. As a baseline, we also include a Q-learning agent using only primitive action, receiving an intrinsic novelty bonus $\beta/\sqrt{n(s,a)}$, with $\beta = 0.01$.

The learning curves are shown in Figure 3. Plots (a) – (d) show the learning curves for our simplest benchmark domains: four rooms (undirected and directed), Rubik's cube end game, and the tower of Hanoi. All option-based agents reach the maximum reward relatively quickly, with the exception of the shortest-path-novelty method, which exhibits slower learning on the tower of Hanoi.

Plots (e) – (g) compare performance on three undirected domains: office, hex, and large maze. Here the proposed method NEO shows a clear advantage over the alternative approaches. By the end of training, NEO agents achieve the maximum reward of 100 on average, while the other methods fail to achieve higher than 40. We also include a variant of NEO that uses only the first (i.e., smoothest) eigenvector, yielding a single option per update interval, which outperforms the related methods. However, it is evident from the evaluations that leveraging multiple eigenvectors provides additional learning benefits beyond this single eigenvector baseline.

In plots (h) – (k), we present results in directed domains. As in the undirected setting, both NEO variants outperform the alternative methods across all directed tasks. The agent's state transition graphs remained weakly connected throughout most of learning, and we obtained no complex eigenvalues in the construction of any option policy for NEO, with all 4 smoothest eigenvalues computed every $H$ update iterations being real valued. We hypothesized that the presence of sink states stabilizes the eigenvalues and makes it more likely that they are real valued. In Appendix E, we provide a theoretical bound showing that the imaginary part of the eigenvalue is notably impacted by the sink weights and present empirical results. For the NYC directed domain, eigen-options could not be computed because the solver would crash. We suspect this issue arises from the interaction between the polar decomposition method and initially weakly connected graphs, which may impair numerical stability during option computation.

In Figure 4, we present the number of unique nodes visited by each method, with various values of the option initialization probability $init_P$. Lower initialization probabilities consistently improve coverage for all methods. However, only NEO approaches full coverage in hex and large maze. These are the largest symmetric domains where it is possible to compare all methods. In these domains, NEO achieves a higher unique-node count during evaluations, roughly twice that of the competing approaches.

Overall, these evaluations demonstrate that NEO achieves or surpasses state-of-the-art performance in both undirected and directed environments, in terms of exploration and, by extension, learning.

**One boundary of the Fiedler vector tends to sit in a well-visited region.** Eigen-options and cover options include the Fiedler vector among their option sets. To understand the performance gap between these methods and NEO, we analyze how the *signed* option pair, corresponding to eigenvectors $f$ and $-f$, can undo progress by steering the agent back to states that are relatively well

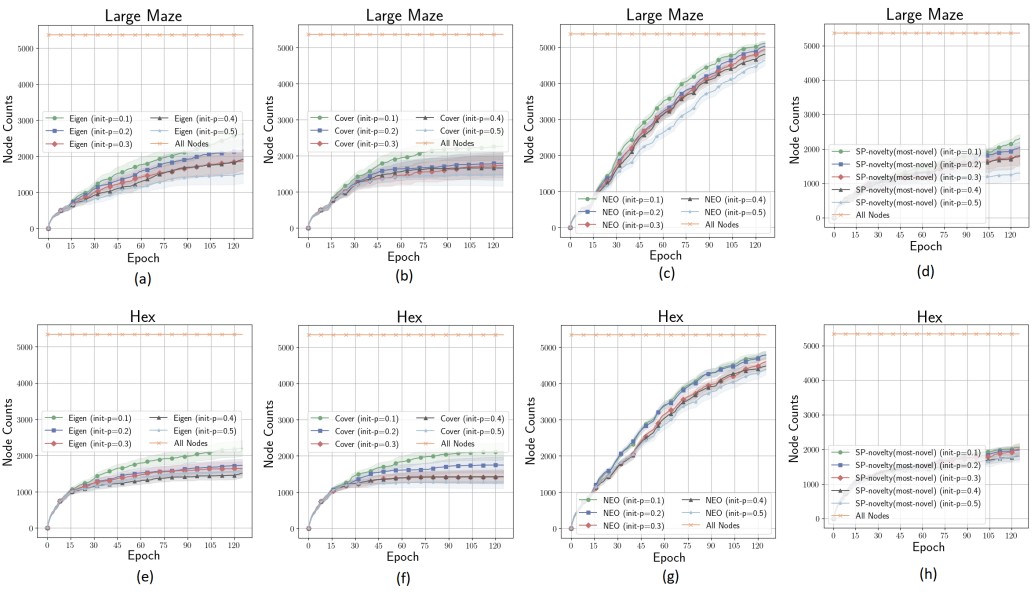

Figure 4: Comparative initialization probability and node count evaluations of learning agents with different option discovery methods across a set of domains.

explored. For this analysis, we use the two largest symmetric domains where Fiedler vectors can be computed, hex and large maze.

Consider an agent that begins near a corner of the full graph and builds the graph incrementally. That corner typically becomes the region that is the most connected and the most frequently visited. Placing one boundary (say $b_1$, an extremum of the Fiedler vector $f$) in this region tends to be part of the first non-trivial solution of the Rayleigh quotient $\mathcal{R}(f) = \frac{\sum_{(u,v)\in E}\left(f_u - f_v\right)^2}{f^\top f}$ because the dense local connectivity allows $f$ to taper off gradually, keeping each edge difference $(f_u - f_v)$ and thus the numerator small. If the second boundary were nearby, neighbouring nodes would carry opposite extreme values, the numerator would spike, and the quotient would rise. Minimising the quotient therefore pushes the second boundary to a geometrically far away state, such as another corner of the graph, rarely visited.

After 250K steps, we freeze the exploration graph and visitation counts $N(\cdot)$, compute eigen-options, and extract the Fiedler vector $f$. Let $b_1$ and $b_2$ be the two boundary states (the extrema of $f$), and define $b_{\text{low}} := \arg\min\{N(b_1), N(b_2)\}$, and $b_{\text{high}} := \arg\max\{N(b_1), N(b_2)\}$. We initialize the agent at $b_{\text{low}}$ and execute the Fiedler-vector option that follows the increasing $f$-gradient, moving toward the opposite boundary until the gradient stalls at a local or global maximum. Our aim is to assess whether one of the pair tends to undo progress. Table 1 shows the visitation counts $N(.)$ at Fiedler vector boundaries $b_{\text{low}}$ and $b_{\text{high}}$, the difference, and the visitation counts at the option termination state. For both domains, one of the signed options lifts the agent from relatively novel states to frequently visited states (e.g., mean $68.5 \to 822.6$, median $14.5 \to 745.0$ in hex). Because the two options are chosen with the same probability, the option that drives the agent back to the high-visitation boundary can directly undo the progress made by its counterpart that moves toward the low-visitation boundary.

Table 1: Visitation count statistics in 20 runs.

| Domain | $N(b_{\text{low}})$ | | $N(b_{\text{high}})$ | | $N(b_{\text{high}}) - N(b_{\text{low}})$ | | $N(\text{termination})$ | |
|---|---|---|---|---|---|---|---|---|
| | **Mean** | **Median** | **Mean** | **Median** | **Mean** | **Median** | **Mean** | **Median** |
| Large Maze | 129.8 | 39.0 | 1355.1 | 1070.5 | 1225.3 | 952.5 | 985.6 | 638.0 |
| Hex | 68.5 | 14.5 | 958.9 | 503.5 | 890.5 | 467.0 | 822.6 | 745.0 |

## 7 Conclusion and Future Work

We presented a principled approach for generating exploration options via a proposed source Laplacian, leveraging theoretical insights and spectral graph properties to guide exploration to novel parts of the state space. The proposed method is shown to drive agents toward novel regions, improving learning performance across a range of challenging domains compared to the existing state-of-the-art. The approach is shown to be applicable to both undirected and directed environments, demonstrating versatility.

Although the theoretical and empirical results presented in this paper center on using state novelty as the driving signal, the proposed framework is general and is not restricted to the use of novelty alone. In fact, any real-valued state functions could be plugged in. Future work can, for instance, replace or augment novelty with expected reward, reward prediction error, or competence signals. We discuss the future extension to symmetric Hilbert spaces in Appendix D.

Recent approaches to approximate the eigenvector of the Laplacian [31, 32, 33] and commute times [34] via these eigenvectors provide a foundation for future work in extending Laplacian-based option discovery methods to the large and continuous setting.

A current limitation is obtaining eigenvectors of the source Laplacian $L_\zeta$ in Hilbert space as done in the symmetric case of the random walk Laplacian [33]. This limitation currently exists for all asymmetric Laplacians, with potentially complex eigenvalues, complex eigenvectors, and non-Euclidean inner products. Another limitation is that exploration policies are currently being learned for a particular environment, one at a time. Future work can explore how these policies can be learnt in a general way, to be used effectively in other parts of the environment or even in new environments, drawing inspiration from approaches to temporal abstraction that focus on generalisation [35].

## Acknowledgements

We would like to thank the members of the Bath Reinforcement Learning Laboratory for their comments and useful suggestions.

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

# Appendix

## A. Experimental Details

**Hardware.** All results are obtained using a 9900k CPU, 16GB of ram, a 256 SSD hard drive and a 2080Ti NVIDIA GPU. Results are obtained in hours on a local computer.

**Parameter settings.**

| Parameter | Value |
|---|---|
| Epoch length | 2,000 steps |
| Option initialization probability ($init_p$) | {0.1, 0.2, 0.3, 0.4, 0.5} |
| Update horizon ($H$) | 5× number of nodes in the domain |
| Learning rate ($\alpha$) | 0.4 |
| Discount factor ($\gamma$) | 0.99 |
| Epsilon ($\varepsilon$) | 0.1 |
| Instances per agent | 20 |

**Domain size.**

| Environment | Number of nodes | Number of edges |
|---|---|---|
| Four rooms | 104 | 168 |
| Office | 2 558 | 3 849 |
| Double large maze | 5 363 | 9 010 |
| Tower of Hanoi | 729 | 1 092 |
| Hex | 5 334 | 8 346 |
| Torus | 5 336 | 19 028 |
| NYC | 10 000 | 22 354 |
| Rubik's cube | 1 051 | 1 380 |

## B. Novelty Scaling Function

As noted in the main text, raw novelty counts are scaled by using Equation 2, which has two parameters, $\delta$ and $k$. Here we present experiments with different values of these parameters in the directed hex domain, shown in Figure 5, where we represent $\delta$ by $d$ in the plots. The experimental conditions are otherwise identical to those reported in Figure 4 of the main paper.

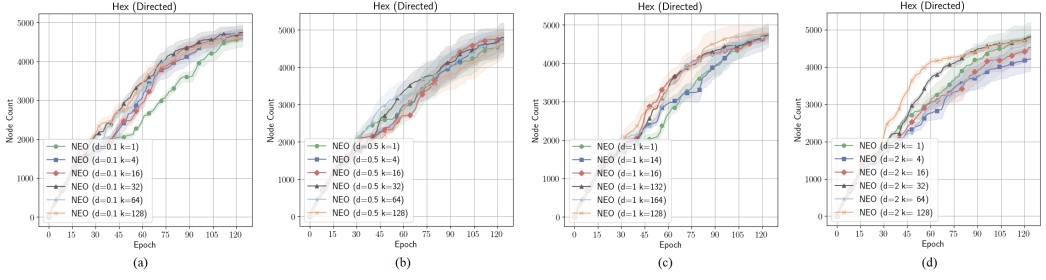

Figure 5: Hyperparameter sweeps of $\delta$ ($d$) and $k$ in the directed hex domain.

## C. Solving Eigen-options and Cover-options with Policy Iteration

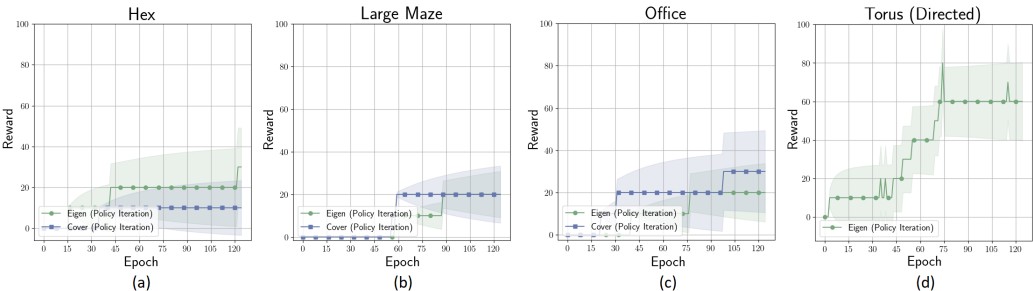

Figure 6: Comparative sparse reward empirical evaluation of the learning agents with the option discovery methods eigen-options and cover-options with option policies solved using policy iteration across a set of domains.

In the main paper, for all eigenvector-based methods, we derived option policies by using the eigenvector values directly, hill-climbing toward higher values. In the original eigen-options work [1], however, option policies are obtained by treating the eigenvector $f$ as a potential function and defining an intrinsic reward $R(s, a, s') = f(s') - f(s)$, then solving for the option policy using policy iteration. Here we report results for the eigen-options and cover-options using this original formulation [1]: we construct the intrinsic reward from the associated eigenvector and run policy iteration until the policy stabilizes (using a threshold 0.001 as in the original work [1]). As shown in Figure 6, solving option policies in this manner does not yield a significant performance improvement against the eigen-option and cover-option results for the set of large domains evaluated in the main paper.

## D. Extension to Continuous Domains

As mentioned in the main paper, if the graph is symmetric, then, for our source Laplacian, we can express the Rayleigh quotient [36] as follows:

$$\mathcal{R}(x) = \frac{x^\top L_\zeta\, x}{x^\top x} = \frac{\frac{1}{2} \sum_{i,j} P_{ij}(x_i - x_j)^2 \;+\; \sum_i \zeta_i\, x_i^2}{\sum_i x_i^2}$$

This is the standard Rayleigh quiotient for a symmetric matrix $L^{rw} = I - P$ with the additional term $\sum_i \zeta_i\, x_i^2$ which is the added squared smoothness error contributions $\zeta_i(x_i - 0)^2$ between the values $x_i$ and the pulls to zero weighted by $\zeta_i$. Therefore, extending the proposed method (NEO) to symmetric continuous domains is in principle, simply a matter of extending currently Laplacian representation approaches such as [33] by adding a mean square error loss to move sink nodes towards zero based on sink strengths which we can obtain by pseudo-count methods such as distribution random network distillation [5].

## E. Bounding $|\operatorname{Im}\lambda(\zeta)|$

We proceed in two stages. First, we show that when $\zeta$ is sufficiently large, the spectrum of our source Laplacian splits into two distinct families, type 1 eigenvalues (those with $|\lambda| \geq \zeta$) and type 2 eigenvalues (those with $|\lambda| < \zeta$), each occupying a different region and characterized by different spectral radii. Secondly, we form the Schur complement [37] of a carefully chosen submatrix of the source Laplacian. By working with an expanded version of this complement, we derive a bound on the imaginary parts of the type 2 eigenvalues. This is achieved by quantifying how the real and imaginary components of the corresponding eigenvectors interact with the Schur complement. Table 2 gives a list of symbols used throughout this section.

| Symbol | Description |
|---|---|
| $n$ | Matrix dimension; indices split into $S$ and $T$ |
| $S, T$ | Index partition with $|S| = m$, $|T| = t$; $S$ precedes $T$ (block order) |
| $P \in \mathbb{R}^{n \times n}$ | Row-stochastic: $p_{ij} \geq 0$, $\sum_j p_{ij} = 1$ |
| $P_{SS}, P_{ST}, P_{TS}, P_{TT}$ | Blocks of $P$ after permutation |
| $\zeta > 0$ | Uniform sink weight applied to $S$ |
| $\Gamma(\zeta)$ | $\mathrm{diag}(\underbrace{\zeta, \ldots, \zeta}_{m}, \underbrace{0, \ldots, 0}_{t})$ |
| $L(\zeta)$ | Source Laplacian $I + \Gamma(\zeta) - P$ |
| $A(\zeta)$ | $(\zeta + 1)I_m - P_{SS}$ (the $SS$ block) |
| $B, C, D$ | $B = -P_{ST}$, $C = -P_{TS}$, $D = I_t - P_{TT}$ |
| $\mathrm{spec}(X)$ | Spectrum (set of eigenvalues) of $X$ |
| $S_\zeta(\lambda)$ | Schur complement $(D - \lambda I) - C(A - \lambda I)^{-1}B$ |
| $K(\lambda)$ | Coupling $C(A - \lambda I)^{-1}B$ |
| $R(\lambda, \zeta)$ | $(P_{SS} + \lambda I)/(\zeta + 1)$ in the Neumann series |
| $R_2(\lambda, \zeta)$ | Tail $\sum_{j \geq 2} R(\lambda, \zeta)^j$ |
| $D_H, D_{SH}$ | Symmetric/skew parts: $D_H = \frac{1}{2}(D + D^\top)$, $D_{SH} = \frac{1}{2}(D - D^\top)$ |
| $y$ | $T$-block (source nodes block) of an eigenvector $\begin{bmatrix} x \\ y \end{bmatrix}$ of $L(\zeta)$ |
| $x$ | $S$-block (sink nodes block) of the same eigenvector |
| $\|\cdot\|_\infty$ | Max absolute row-sum operator norm |
| $\|\cdot\|_2$ | Spectral (Euclidean) operator norm |

Table 2: List of symbols.

## Standard inequalities and tools

Operator norms and parts: We use $\|\cdot\|_\infty$ to denote the maximum absolute row-sum, $\|\cdot\|_2$ the spectral norm. For a (possibly complex) matrix $M$, we write $M_H = \frac{1}{2}(M + M^*)$ and $M_{SH} = \frac{1}{2i}(M - M^*)$. For a real matrix, $M = M_H + M_{SH}$ is the split into symmetric and skew-symmetric parts.

Determinant factorization: For a block matrix $\begin{bmatrix} E & F \\ G & H \end{bmatrix}$ with $E$ invertible,

$$\det \begin{bmatrix} E & F \\ G & H \end{bmatrix} = \det(E) \cdot \det(H - GE^{-1}F),$$

where $H - GE^{-1}F$ is known as the *Schur complement* [37].

Neumann-series resolvent: If $\|R\| < 1$ in an operator norm, then $(I - R)^{-1} = \sum_{j=0}^{\infty} R^j$ converges and $\|(I - R)^{-1}\| \leq 1/(1 - \|R\|)$.

Cauchy–Schwarz Inequality: Given two vectors $u$ and $v$, $|u^*v| \leq \|u\|_2 \|v\|_2$.

## Source Laplacian and block form

Fix $n \geq 2$ and a nonempty $T \subset \{1, \ldots, n\}$ with $|T| \geq 1 = t$ and $S := \{1, \ldots, n\} \setminus T$ with $|S| = m$. For $\zeta > 0$ define
$$\Gamma(\zeta) := \mathrm{diag}(\underbrace{\zeta, \ldots, \zeta}_{m \text{ indices in } S}, \underbrace{0, \ldots, 0}_{t \text{ indices in } T}).$$

Here we consider the T-block as the source nodes and the S-block as the sink nodes, with sink strength $= \zeta$.

Let $P \in \mathbb{R}^{n \times n}$ be row-stochastic. Set $L(\zeta) := I_n + \Gamma(\zeta) - P$. After ordering indices so

$S$ comes first,

$$L(\zeta) = \begin{bmatrix} A(\zeta) & B \\ C & D \end{bmatrix}, \qquad A(\zeta) := (\zeta+1)I_m - P_{SS}, \ B := -P_{ST}, \ C := -P_{TS}, \ D := I_t - P_{TT}. \tag{3}$$

Notice that the sink-strength $\zeta$ only affects the $SS$ block $A(\zeta)$: it appears through the added diagonal damping on the sink-indexed nodes and therefore modifies only the interactions among sinks. The remaining blocks are $\zeta$-independent: $B$ contains connections from sink nodes to source nodes, $C$ contains connections from source nodes back to sink nodes, and $D$ contains connections among the source nodes themselves.

**Schur complement and the $T$-block equation**

For $\lambda \in \mathbb{C}$,

$$L(\zeta) - \lambda I_n = \begin{bmatrix} A(\zeta) - \lambda I_m & B \\ C & D - \lambda I_t \end{bmatrix}.$$

Whenever $A(\zeta) - \lambda I_m$ is invertible, we can define the Schur complement [37] as:

$$S_\zeta(\lambda) := (D - \lambda I_t) - C\,(A(\zeta) - \lambda I_m)^{-1} B. \tag{4}$$

Then the block determinant identity gives

$$\det(L(\zeta) - \lambda I_n) = \det(A(\zeta) - \lambda I_m) \cdot \det S_\zeta(\lambda). \tag{5}$$

Moreover, writing an eigenvector of $L(\zeta)$ in blocks as $\begin{bmatrix} x \\ y \end{bmatrix}$, where y is the T-block (source node block) of an eigenvector and x is the S-block (sink node block) of the eigenvector, the equation $(L(\zeta) - \lambda I)\begin{bmatrix} x \\ y \end{bmatrix} = 0$ is equivalent to

$$(A(\zeta) - \lambda I_m)x + By = 0, \qquad Cx + (D - \lambda I_t)y = 0.$$

If $A(\zeta) - \lambda I_m$ is invertible, then

$$x = -(A(\zeta) - \lambda I_m)^{-1} By, \tag{6}$$

and substituting into the second block gives

$$S_\zeta(\lambda)\,y = 0. \tag{7}$$

Thus we focus on $y$ because it is the $T$-block of the full eigenvector and it determines $x$ via (6).

**Type-II region inside $|\lambda| < \zeta$ and invertibility of $A(\zeta) - \lambda I$**

**Lemma 7.1** (Resolvent exclusion). Let $A(\zeta) = (\zeta + 1)I_m - P_{SS}$ with $P$ row-stochastic. If $|\lambda| < \zeta$, then $A(\zeta) - \lambda I_m$ is invertible. Hence no eigenvalue of $A(\zeta)$ lies in $\{|\lambda| < \zeta\}$.

*Proof.* Factor

$$A(\zeta) - \lambda I_m = (\zeta + 1)\left(I_m - \tfrac{P_{SS} + \lambda I_m}{\zeta + 1}\right).$$

Let $Q := (P_{SS} + \lambda I_m)/(\zeta + 1)$. Since $P$ is row-stochastic, $\|P_{SS}\|_\infty \le 1$, so

$$\|Q\|_\infty \le \frac{\|P_{SS}\|_\infty + |\lambda|}{\zeta + 1} \le \frac{1 + |\lambda|}{\zeta + 1} < 1 \qquad (|\lambda| < \zeta).$$

Thus $I_m - Q$ is invertible by the Neumann series, hence so is $A(\zeta) - \lambda I_m$. $\qquad \square$

**Remark 7.1** (Type-II eigenvalues). Inside the disk $|\lambda| < \zeta$, Lemma 7.1 guarantees $A(\zeta) - \lambda I_m$ is invertible. Therefore any eigenvalue of $L(\zeta)$ with $|\lambda| < \zeta$ must satisfy $\det S_\zeta(\lambda) = 0$. We call such eigenvalues *type-II*.

**Neumann expansion in the smaller disk $|\lambda| \leq r\zeta$**

Fix $0 < r < 1$ and restrict to $|\lambda| \leq r\zeta$. Define

$$R(\lambda, \zeta) := \frac{P_{SS} + \lambda I_m}{\zeta + 1}, \qquad R_2(\lambda, \zeta) := \sum_{j=2}^{\infty} R(\lambda, \zeta)^j.$$

For clean, uniform 2-norm bounds on this disk, we assume the *gap condition*

$$\delta := (\zeta + 1) - (\|P_{SS}\|_2 + r\zeta) > 0. \tag{8}$$

(Equivalently: $\|P_{SS}\|_2 + r\zeta < \zeta + 1$.)

**Lemma 7.2** (Neumann-series resolvent with tail bound). Assume (8) and $|\lambda| \leq r\zeta$. Then $\|R(\lambda, \zeta)\|_2 < 1$ and

$$(A(\zeta) - \lambda I_m)^{-1} = \frac{1}{\zeta + 1}\Big(I_m + R(\lambda, \zeta) + R_2(\lambda, \zeta)\Big), \tag{9}$$

and the tail satisfies the *uniform* bound

$$\|R_2(\lambda, \zeta)\|_2 \leq \frac{\|R(\lambda, \zeta)\|_2^2}{1 - \|R(\lambda, \zeta)\|_2} \leq \frac{(\|P_{SS}\|_2 + r\zeta)^2}{(\zeta + 1)\,\delta}. \tag{10}$$

*Proof.* First we have:
$$A(\zeta) - \lambda I_m = (\zeta + 1)\big(I_m - R(\lambda, \zeta)\big). \tag{11}$$

Next since $\|R(\lambda, \zeta)\|_2 < 1$,

$$\|R(\lambda, \zeta)\|_2 \leq \frac{\|P_{SS}\|_2 + |\lambda|}{\zeta + 1} \leq \frac{\|P_{SS}\|_2 + r\zeta}{\zeta + 1} = 1 - \frac{\delta}{\zeta + 1} < 1,$$

so $(I_m - R)^{-1} = \sum_{j=0}^{\infty} R^j$ and (9) follows. Finally,

$$\|R_2\|_2 = \Big\|\sum_{j=2}^{\infty} R^j\Big\|_2 \leq \sum_{j=2}^{\infty} \|R\|_2^j = \frac{\|R\|_2^2}{1 - \|R\|_2}.$$

Using $1 - \|R\|_2 \geq 1 - \frac{\|P_{SS}\|_2 + r\zeta}{\zeta + 1} = \frac{\delta}{\zeta + 1}$ yields (10). $\qquad\square$

**Theorem 7.3** (Schur expansion). Assume (8) and $|\lambda| \leq r\zeta$. Then

$$C(A(\zeta) - \lambda I_m)^{-1}B = \frac{CB}{\zeta + 1} + \frac{C(P_{SS} + \lambda I_m)B}{(\zeta + 1)^2} + \frac{C\,R_2(\lambda, \zeta)\,B}{\zeta + 1}, \tag{12}$$

$$S_\zeta(\lambda) = (D - \lambda I_t) - \frac{CB}{\zeta + 1} + E_\zeta(\lambda), \tag{13}$$

where

$$E_\zeta(\lambda) := -\frac{C(P_{SS} + \lambda I_m)B}{(\zeta + 1)^2} - \frac{C\,R_2(\lambda, \zeta)\,B}{\zeta + 1} \tag{14}$$

$$\tag{15}$$

*Proof.* Expand $C(A(\zeta) - \lambda I_m)^{-1}B$ by inserting (9):

$$C(A(\zeta) - \lambda I_m)^{-1}B = \frac{1}{\zeta + 1} C\Big(I_m + R(\lambda, \zeta) + R_2(\lambda, \zeta)\Big)B. \tag{16}$$

$$= \frac{1}{\zeta + 1}\Big(CI_mB + CR(\lambda, \zeta)B + CR_2(\lambda, \zeta)B\Big). \tag{17}$$

$$= \frac{CB}{\zeta + 1} + \frac{C(P_{SS} + \lambda I_m)B}{(\zeta + 1)^2} + \frac{CR_2(\lambda, \zeta)B}{\zeta + 1} \tag{18}$$

Then the Schur complement expands as:

$$S_\zeta(\lambda) = (D - \lambda I_t) - C(A(\zeta) - \lambda I_m)^{-1}B. \tag{19}$$

$$= (D - \lambda I_t) - \left[\frac{CB}{\zeta + 1} + \frac{C(P_{SS} + \lambda I_m)B}{(\zeta + 1)^2} + \frac{CR_2(\lambda, \zeta)B}{\zeta + 1}\right] \tag{20}$$

$$\tag{21}$$

$$= (D - \lambda I_t) - \frac{CB}{\zeta + 1} + E_\zeta(\lambda) \tag{22}$$

giving (13)–(14), with $E_\zeta(\lambda)$ being a collection of the higher order terms of (20).

$\square$

**Lemma 7.4** (Envelope bound on $K(\lambda)$ in $|\lambda| \leq r\zeta$)**.** Let $K(\lambda) := C(A(\zeta) - \lambda I_m)^{-1}B$. Assume (8) and $|\lambda| \leq r\zeta$. Then

$$\|K(\lambda)\|_2 \ \leq\ \frac{\|B\|_2\,\|C\|_2}{\delta} \qquad \Big(\delta = (\zeta + 1) - (\|P_{SS}\|_2 + r\zeta)\Big).$$

*Proof.* From Lemma 7.2,

$$\|(A(\zeta) - \lambda I_m)^{-1}\|_2 = \frac{1}{\zeta + 1}\|(I_m - R)^{-1}\|_2 \leq \frac{1}{\zeta + 1} \cdot \frac{1}{1 - \|R\|_2} \leq \frac{1}{\zeta + 1} \cdot \frac{1}{\delta/(\zeta + 1)} = \frac{1}{\delta}.$$

Therefore

$$\|K(\lambda)\|_2 \leq \|C\|_2\,\|(A(\zeta) - \lambda I_m)^{-1}\|_2\,\|B\|_2 \leq \frac{\|B\|_2\|C\|_2}{\delta}.$$

$\square$

**Expanded imaginary-part bound in $|\lambda| \leq r\zeta$**

**Theorem 7.5** (Explicit imaginary-part bound (disk $|\lambda| \leq r\zeta$))**.** Fix $0 < r < 1$ and assume (8). Let $\lambda(\zeta) \in \operatorname{spec} L(\zeta)$ satisfy $|\lambda(\zeta)| \leq r\zeta$. Since $r\zeta < \zeta$, Lemma 7.1 implies $A(\zeta) - \lambda(\zeta)I_m$ is invertible. Let $\begin{bmatrix} x \\ y \end{bmatrix} \neq 0$ be a corresponding eigenvector of $L(\zeta)$, written in $S/T$ blocks. Then $y \neq 0$ satisfies the Schur-complement eigenvector equation

$$S_\zeta(\lambda(\zeta))\,y = 0,$$

and we can assume a normalized $y$ so that $\|y\|_2 = 1$.

Split $D = D_H + D_{SH}$ with $D_H = \frac{1}{2}(D + D^\top)$ and $D_{SH} = \frac{1}{2}(D - D^\top)$. Because $D_{SH}^\top = -D_{SH}$ (real skew-symmetric), the scalar $y^*D_{SH}y$ is purely imaginary; hence there exists $\alpha(\zeta) \in \mathbb{R}$ such that

$$y^*D_{SH}y = i\,\alpha(\zeta), \qquad \beta(\zeta) := |\alpha(\zeta)|.$$

Define $a := \|P_{SS}\|_2 + r\zeta$ and $\delta$ as in (8). Then

$$|\Im\lambda(\zeta)| \ \leq\ \beta(\zeta) + \frac{\|B\|_2\,\|C\|_2}{\zeta + 1} + \frac{\|B\|_2\,\|C\|_2}{(\zeta + 1)^2}\left(a + \frac{a^2}{\delta}\right).$$

*Proof.* From $S_\zeta(\lambda)y = 0$ and the expansion (13),

$$0 = y^*\left[(D - \lambda I_t) - \frac{CB}{\zeta + 1} + E_\zeta(\lambda)\right]y = y^*(D - \lambda I_t)y - \frac{1}{\zeta + 1}y^*CBy + y^*E_\zeta(\lambda)y. \tag{P1}$$

Write $y^*Dy = y^*D_Hy + y^*D_{SH}y = y^*D_Hy + i\alpha(\zeta)$ and take imaginary parts of (P1):

$$|\Im\lambda(\zeta)| \leq |\alpha(\zeta)| + \frac{|y^*CBy|}{\zeta + 1} + |y^*E_\zeta(\lambda)y|. \tag{P2}$$

(i) Since $y^*CBy = (By)^*(Cy)$, Cauchy–Schwarz gives

$$|y^*CBy| \leq \|By\|_2\|Cy\|_2 \leq \|B\|_2\|C\|_2.$$

(ii) Using (14), $\|y\|_2 = 1$, $|y^* M y| \le \|M\|_2$, and submultiplicativity,

$$|y^* E_\zeta(\lambda) y| \le \|B\|_2 \|C\|_2 \left( \frac{\|P_{SS}\|_2 + |\lambda|}{(\zeta + 1)^2} + \frac{\|R_2(\lambda, \zeta)\|_2}{\zeta + 1} \right).$$

With $|\lambda| \le r\zeta$ this is

$$|y^* E_\zeta(\lambda) y| \le \|B\|_2 \|C\|_2 \left( \frac{a}{(\zeta + 1)^2} + \frac{\|R_2(\lambda, \zeta)\|_2}{\zeta + 1} \right).$$

Finally, apply the tail bound (10):

$$\frac{\|R_2(\lambda, \zeta)\|_2}{\zeta + 1} \le \frac{1}{\zeta + 1} \cdot \frac{a^2}{(\zeta + 1)\delta} = \frac{a^2}{(\zeta + 1)^2 \delta}.$$

So

$$|y^* E_\zeta(\lambda) y| \le \frac{\|B\|_2 \|C\|_2}{(\zeta + 1)^2} \left( a + \frac{a^2}{\delta} \right).$$

Insert (i) and (ii) into (P2) and use $\beta(\zeta) = |\alpha(\zeta)|$. $\qquad \square$

**Remark 7.2.** If $D$ is symmetric, then $D_{SH} = 0$ so $\beta(\zeta) = 0$, and the imaginary part is controlled entirely by the coupling terms that decay with $\zeta$ (through factors $(\zeta + 1)^{-1}$ and $(\zeta + 1)^{-2}$) provided the gap $\delta = (\zeta + 1) - (\|P_{SS}\|_2 + r\zeta)$ stays positive.

**Eigenvalue bound observations**

Figure 7 presents nine plots showing the real versus imaginary parts of the eigenvalues of our source Laplacian with a sink strength $\zeta$. According to our theory, the sink strength $\zeta$ imposes an upper bound on the imaginary parts of the type-2 eigenvalues. To test this, we work on the 104-node directed four rooms domain, in which four nodes are designated as sources with sink-strength zero and the remaining 100 nodes as sinks with strength $\zeta$). For each of the nine chosen values of $\zeta$, we repeat the following 5,000 times: randomly select four source nodes, assign sink-strengths on the remaining 100 nodes, build the directed Laplacian, compute its spectrum, and collect all eigenvalues.

As $\zeta$ increases, the two classes of eigenvalues – type 1 (those farther from zero) and type 2 (those clustering near zero) – separate into distinct regions. In particular, the imaginary parts of the type 2 eigenvalues shrink toward zero as $\zeta$ grows. When $\zeta = 0$ (in plot a), our Laplacian reduces to the standard random-walk form where there is no distinction between type 1 and type 2 eigenvalues, with both having significant imaginary parts. By contrast, at the largest $\zeta$ (plot i), only the type 1 eigenvalues that exceed exhibit significant nonzero imaginary parts.

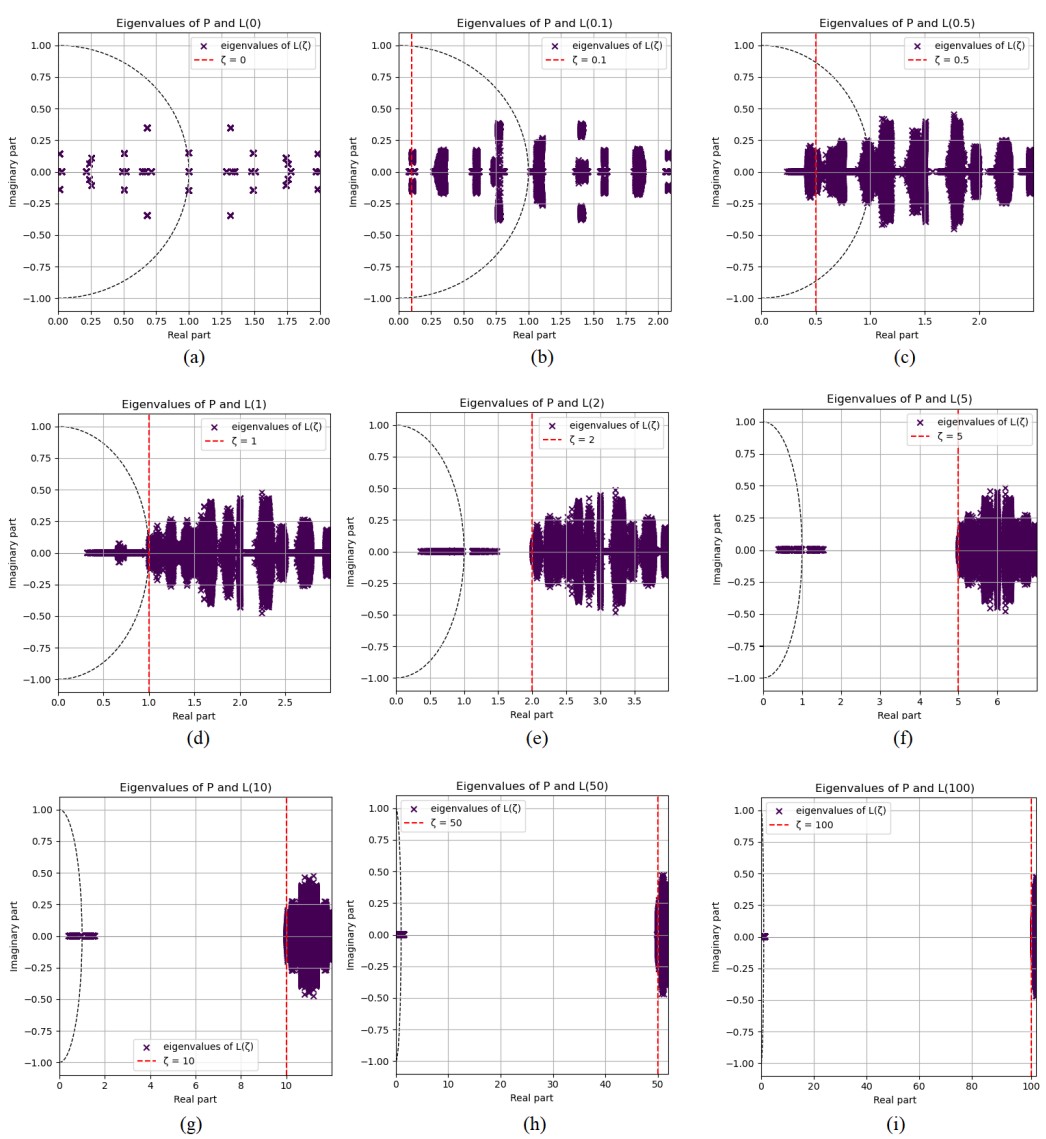

Figure 7: Eigenvalue plots, where we show the real and imaginary parts for source Laplacian under different sink strengths.

