# OpenReview forum: "Novel Exploration via Orthogonality"
_NeurIPS.cc/2025/Conference — NeurIPS 2025 poster_

### Official Review · Reviewer_jL2w · 2025-07-02

**Clarity:** 2
**Significance:** 3
**Originality:** 4
**Rating:** 5
**Confidence:** 4

**Summary:**

The authors introduce Novel Exploration via Orthogonality (NEO), a method to improve exploration in Reinforcement Learning (RL) through option discovery based on spectral graph properties. NEO builds on Laplacian-based approaches by incorporating sink-weights into the graph Laplacian, where sink weights are derived from state visitation counts to encode novelty. The resulting modified Laplacian yields eigenvectors whose gradients guide the agent toward novel regions of the state space. Unlike previous work, NEO extends naturally to both undirected and directed graphs, supported by theoretical guarantees regarding eigenvector properties and gradient flows. The method is evaluated in several RL environments, showing that NEO outperforms existing option discovery baselines, including eigenoptions and cover options.

**Questions:**

- The experiments are performed exclusively in deterministic settings, while the theoretical formulation of the MDP in the Foundations Section allows for stochastic transitions $\mathcal{T}(s,a,s’)$. How robust is NEO to non-deterministic environments, where edge weights in the empirical graph reflect averaged, probabilistic transitions rather than deterministic ones?
- Assumptions in Theorem 3.2: From my understanding, the proof relies on $\lambda < \max_i \gamma_i$, which implies $\gamma_i > \lambda$ only under certain conditions, such as a uniform sink set. Additionally, to establish the maximality argument, the existence of at least one source node appears necessary. Could the authors clarify these assumptions explicitly, or correct any misunderstanding?
- Have the authors considered applying NEO in deep RL environments, where state spaces are continuous and high-dimensional? Even conceptual discussion or preliminary experiments would be valuable to understand potential extensions.
- The construction of sink weights involves a scaling function $F(.)$ with hyperparameters $\delta$ and $k$. How sensitive is NEO’s performance to these choices? Can these be linked to the assumptions that eigenvalues $\lambda$ exists where $N(i) > \lambda$?
- Computing eigenvectors for large graphs can be computationally expensive. Can the authors provide runtime comparisons or estimates of computational overhead relative to baselines?

**Ethical Concerns:**

["NO or VERY MINOR ethics concerns only"]

**Final Justification:**

The rebuttal addressed my main concerns regarding robustness in stochastic environments, assumptions in Theorem 3.2, sensitivity to sink-weight scaling, and runtime measurements, providing clear explanations and additional experiments. The rationale for using cosine similarity in high-dimensional experiments was well explained, though I still believe future work could benefit from applying coverage- or reward-based evaluations in more complex, non-tabular domains. Overall, the combination of theoretical contributions, extensive experiments, and satisfactory rebuttal responses outweigh the remaining minor limitations, and I have accordingly increased my score.

**Limitations:**

yes

**Paper Formatting Concerns:**

See clarity weaknesses

**Quality:**

3

**Strengths And Weaknesses:**

1. Quality:
    - Strengths:
        - The paper presents a technically sound approach, building on solid spectral graph theory foundations.
        - The paper is generally well structured and well written
        - The theoretical contributions are well-developed, including $\lambda$-bounds, gradient guarantees, and proofs of eigenvector properties.
        - The empirical evaluation is extensive, covering eight domains, both directed and undirected.
    - Weaknesses:
        - While the theoretical results are compelling, the practical implications for large, continuous domains are not fully addressed.
        - Most of the empirical evaluation focuses primarily on discrete, synthetic environments.
2. Clarity:
    - Strengths:
        - The formalism around the Source Laplacian and the construction of the NEO options is clear and mathematically well-structured.
        - Algorithm 1 is helpful in understanding the practical implementation of NEO.
    - Weaknesses:
        - In Figure 3, inconsistent coloring for baselines makes comparisons, particularly between directed and undirected versions of the same domain, unnecessarily difficult.
        - Theorem 3.3 suffers from overloaded notation where $i$ is the index of the eigenvector value $f_i$ with $N(i) > \lambda$ and nodes along a constructed path  $i = v_0 \rightarrow \dots \rightarrow v_k$, which can confuse readers
        - Minor but noticeable formatting and typographical issues detract from polish, including:
            - State action pair $(s,a)$ (l.27)
            - Missing spacing for $\gamma$ in (l. 95)
            - max(f(.)) in (l.140)
            - Buggy reference in (l.216)
        - References to baseline models in Evaluation setup (l. 210 - 220) are missing which hinders reproducability
3. Significance:
    - Strengths:
        - The empirical results suggest that NEO significantly improves learning efficiency, especially in directed and large-scale domains.
        - Results seem impactful with NEO outperforming most other approaches in the evaluation
        - Baselines in the evaluation seem well selected
    - Weaknesses:
        - The evaluation is limited to tabular, synthetic domains. Testing on more complex environments (e.g., Atari) would strengthen the practical impact.
4. Originality:
    - Strengths:
        - The introduction of a modified Laplacian with visitation-based sink weights is a novel and meaningful conceptual contribution.
        - Influences this work is based on are referenced appropriately.

---

> ### Author Rebuttal · Authors · 2025-07-31
>
> We thank the reviewer for their careful reading of our paper and their thoughtful questions.
>
> # Q1. Robustness to stochastic transitions
>
> *Reviewer.*  How robust is NEO to non-deterministic environments, where edge weights in the empirical graph reflect averaged, probabilistic transitions rather than deterministic ones?
>
> **Response.**
>
>  **NEO** shows similar performance improvements in non-deterministic environments, with the caveat that stochasticity makes exploration harder for any agent (even an optimal policy). To showcase robustness we consider our largest domain, the New York City street graph (10,000 states), with injected “wind” noise of probability 1/4 that the agent moves up regardless of its chosen action. Using the transition matrix augmented with this noise, **NEO** continues to outperform related methods, achieving roughly twice the coverage (unique nodes visited) in a reward-free evaluation. The results are below. We also include Four-Rooms to show a small domain example:
>
> | Domain     | Steps   | Method             | Visited / Total | Coverage % |
> | ---------- | ------- | ------------------ | --------------: | ---------: |
> | NYC        | 250,000 | NEO                |  9,224 / 10,000 |      92.24 |
> | NYC        | 250,000 | SP-Novelty         |  4,566 / 10,000 |      45.66 |
> | NYC        | 250,000 | Q-learning-novelty |  3,217 / 10,000 |      32.17 |
> | NYC        | 250,000 | Eigen-options      |  3,637 / 10,000 |      36.37 |
> | Four-rooms | 10,000  | NEO                |       104 / 104 |     100.00 |
> | Four-rooms | 10,000  | SP-Novelty         |       104 / 104 |     100.00 |
> | Four-rooms | 10,000  | Q-learning-novelty |        53 / 104 |      50.96 |
> | Four-rooms | 10,000  | Eigen-options      |        65 / 104 |      62.50 |
>
>
> *Note: Results are averaged over 20 agent instances; all other settings match the non-stochastic setup used in the paper.*
>
> ---
>
> # Q2. Assumptions in Theorem 3.2
>
> *Reviewer.* Assumptions in Theorem 3.2: From my understanding, the proof relies on $\lambda < \max_i \gamma_i$
> , which implies $\gamma_i > \lambda$
>  only under certain conditions, such as a uniform sink set. Additionally, to establish the maximality argument, the existence of at least one source node appears necessary. Could the authors clarify these assumptions explicitly, or correct any misunderstanding?
>
> **Response.** Thank you for the careful reading. The proof uses the bound \$|1+\gamma\_i-\lambda|>1\$ at each sink, so we amend the assumption from \$\lambda<\max\_i\gamma\_i\$ to the minimal \$\lambda<\min\_{i:\gamma\_i>0}\gamma\_i\$. We also assume there is at least one source node \$(\exists i:\gamma\_i=0)\$; this is already implicit in Theorem 3.3 because novelty counts start at \$0\$ \$(\Gamma\_{ii}=N(i)\in\mathbb{Z}\_{\ge 0})\$, so some node has \$N(i)=0\$. With these minimal clarifications, the maximality step follows: no sink attains \$\max\_i |f\_i|\$, so any maximizer is a source.
>
> ---
> # Q3. Deep RL extension (continuous, high-dimensional)
>
> *Reviewer.* Have the authors considered applying NEO in deep RL environments, where state spaces are continuous and high-dimensional? Even conceptual discussion or preliminary experiments would be valuable to understand potential extensions.
>
> **Response.** Yes. we can show that **NEO** extends to high-dimensional and continuous settings.
> To extend to the high-dimensional setting, we can amend the smoothness loss from proper Laplacian representation [2] method
> with an additional smoothness term to zero weight by our sink strengths:
>
> $L_{\text{approx}}(x) =\tfrac12\sum_{ij} P_{ij}(x_i-x_j)^2 +\sum_i \big((\gamma_i x_i - 0\big)^2,$
>
> i.e., the standard \$(x\_i-x\_j)^2\$ smoothness plus a shrinkage \$\big(\gamma\_ix\_i\big)^2\$, with \$\gamma\_i\$ derived from our sink-weight scaling (e.g., novelty).
> We discuss this extension in the appendix and how it's based on the Rayleigh quotient for our source Laplacian:
> That is, If the graph is symmetric, then
>
> $$
> R(x)=\frac{x^\top L x}{x^\top x}
> =\frac{\tfrac12 \sum_{ij} P_{ij}(x_i-x_j)^2 + \sum_i \gamma_i x_i^2}{\sum_i x_i^2},
> $$
>
> which is the standard Rayleigh quotient for \$L=I-P\$ (with \$P\$ symmetric) plus the diagonal term \$\sum\_i \gamma\_i x\_i^2\$, equivalently the squared smoothness-to-zero contributions at sinks.
> For the experiments requested, we compare oracle **NEO** eigenpairs to CNN-based approximations on pixel grids, using visitation weights from the analytically computed hitting time from the start state \$(0,0)\$ in each experiment. We report cosine similarity for the 11 *smoothest* eigenvectors, averaged over 10 random seeds. These results are solely to demonstrate *extendibility*.
>
> *Why not Atari?* Atari is predominantly *directed* and lacks oracle eigenpairs, so cosine agreement cannot be validated; the pixel grids provide ground truth and are currently the most complex domain where approximated eigenvectors learned end-to-end with neural networks have been compared correctly to oracle eigenvectors.
>
> **Cosine similarity (11 smoothest; mean over 10 seeds).**
>
> | Environment | Mean      | Std       | Median    | Min       | Max       | \$\ge 0.99\$ (count/11) |
> | ----------- | --------- | --------- | --------- | --------- | --------- | ----------------------- |
> | GRIDMAZE-19 | 0.994137  | 0.012108  | 0.9984306 | 0.9563711 | 0.9993445 | 10                      |
> | GRIDMAZE-16 | 0.983047  | 0.025448  | 0.9931449 | 0.9075332 | 0.9966728 | 8                       |
> | GridRoom-4  | 0.988293  | 0.010844  | 0.9916129 | 0.9659867 | 0.9993791 | 6                       |
> | GridRoom-1  | 0.9546429 | 0.0689298 | 0.979806  | 0.7444355 | 0.9983348 | 3                       |
>
> **Per-eigenvector cosine similarities (\$v\_1\$–\$v\_{11}\$); values are means over 10 seeds.**
>
> |                 | \$v\_1\$   | \$v\_2\$   | \$v\_3\$  | \$v\_4\$   | \$v\_5\$   | \$v\_6\$   | \$v\_7\$   | \$v\_8\$   | \$v\_9\$   | \$v\_{10}\$ | \$v\_{11}\$ |
> | --------------- | ---------- | ---------- | --------- | ---------- | ---------- | ---------- | ---------- | ---------- | ---------- | ----------- | ----------- |
> | **GRIDMAZE-19** | 0.9563711  | 0.99921966 | 0.9993445 | 0.9985542  | 0.9991034  | 0.9993152  | 0.9984306  | 0.99785894 | 0.99696004 | 0.99832886  | 0.9920205   |
> | **GRIDMAZE-16** | 0.9823314  | 0.9956034  | 0.9962385 | 0.9966582  | 0.9966728  | 0.996615   | 0.99314487 | 0.9917789  | 0.9909922  | 0.96594334  | 0.90753317  |
> | **GridRoom-4**  | 0.9767812  | 0.99859035 | 0.9993791 | 0.99753225 | 0.97569335 | 0.99685365 | 0.9916129  | 0.9977458  | 0.9863626  | 0.98468316  | 0.9659867   |
> | **GridRoom-1**  | 0.96239364 | 0.94523597 | 0.9983348 | 0.96192765 | 0.94789433 | 0.9976875  | 0.98534834 | 0.9970977  | 0.98091066 | 0.979806    | 0.7444355   |
>
> In the experiments we use all the parameters from the original proper laplacian representation paper for the pixel based CNN tasks, and only add the sink weights to pull eigenvector values closer to zero across the spectrum.
> Directed domains are the critical next step. Classical “directed” Laplacian approaches [4] hinge on the stationary distribution of a directed random walk; in high-dimensional continuous spaces, this is *intractable to obtain without first approximating the geometry to a significant degree, making such methods circular in their intractability. Moreover, any Hermitian method (e.g., stationary-distribution–weighted[4] or magnetic variants[5]) is symmetric by construction and thus cannot fully encode directedness. Truly non-Hermitian formulations preserve direction but introduce complex spectra; handling complex-valued eigenfunctions in learning pipelines (inner products, orthogonality/normalization, and potentially complex-valued networks) is highly nontrivial. Our contribution makes progress: we provide gradient guarantees even if eigenvectors are complex; empirically, our eigenpairs are real and the imaginary part of eigenvalues decreases as \$\beta\$ increases; and we outline source/sink option designs that encourage real spectra. We see these as concrete steps for future work toward a principled method to obtain useful eigenvectors of directed Laplacians in Hilbert space.
>
> ---
>
> # Q4. Sensitivity to sink-weight scaling
>
> *Reviewer.* How sensitive is NEO’s performance to these choices? Can these be linked to the assumptions that eigenvalues
>  exists where N(i) > $\lambda$
> ?
>
> **Response.**
> In the appendix (Figure 3) we report a sweep over the hyperparameters δ and k. Across a wide range of values, performance is stable and typically matches the default configuration. Moreover, the method outperforms the competing baselines across hyperparameters, indicating that our approach is not unduly sensitive to δ or k. For a high scaling of k, we form a nearly uniform value for our less novel states and the states at 0 would act as sources - in this sense, we can link certain scaling of k to our first $\lambda-bound$ $0< R(1) < \gamma$ in theorem 3.1.
>
> ---
>
> # Q5. Runtime and computational overhead
>
> *Reviewer.* Can the authors provide runtime comparisons or estimates of computational overhead relative to baselines?
>
> **Response**
> For a fair comparison, we use the large maze environment, which has a total of 5363 nodes and 9010 edges and compute the option eigenvectors or path distances on the full graph:
> | Method       | Time (s)  |
> |--------------|----------:|
> | SP-Novelty   | 1.387825  |
> | NEO          | 3.633035  |
> | Eigenoptions | 6.054768  |
> | Covertime    | 41.021344 |
> Note that the eigenvector methods produce 4 options and the SP-novelty produces one.
>
> [3] Wang et al. Reachability-Aware Laplacian Representation in Reinforcement Learning. arXiv 2022.
>
> [4] Chung. Laplacians and the Cheeger Inequality for Directed Graphs. Annals of Combinatorics 2005.
>
> [5] Zhang et al. MagNet: A Neural Network for Directed Graphs. NeurIPS 2021.

---

> > ### Comment · Reviewer_jL2w · 2025-08-05
> >
> > **Q1.** Thank you for providing the additional experiments for stochastic environments. These experiments show a similar coverage behavior to deterministic environments, which helps clarify my concerns. However, could you please further clarify how the option policy is constructed in these non-deterministic environments (as mentioned on line 10 in Algorithm 1)? Specifically, are the probabilities for different $(s, a, s')$ pairs cached in the graph, and is the action with the highest probability of transitioning to $s'$ selected?
> >
> > **Q2.** The additional clarification regarding the bound is much appreciated. With this, the proof now appears sound to me.
> >
> > **Q3.** Thank you again for providing the additional results. However, I would appreciate some further clarification on why you chose to use the cosine similarity of EVs when compared to CNN-based approximations, as opposed to the reward-based and coverage-based evaluation metrics presented in the paper (Figure 3 & 4). Additionally, I still believe an analysis of the discovered options in an Atari (or similar non-tabular) environment, similar to the approach in the Eigenoptions paper [1], would have been valuable and could have further strengthened the paper.
> >
> > **Q4.** The figure has effectively clarified the sensitivity, and I have no further questions on this matter.
> >
> > **Q5**. For the final question, I once again thank you for the additional experiments, which indicate that NEO performs well in terms of runtime when compared to other baselines. However, could you please clarify what exactly is being measured in terms of time? Is it the time for an entire run, one epoch, or just the time for a single option discovery call?
> >
> > [1] Machado, Marlos C., Marc G. Bellemare, and Michael Bowling. "A laplacian framework for option discovery in reinforcement learning." International Conference on Machine Learning. PMLR, 2017.

---

> ### Author Response · Authors · 2025-08-08
>
> **Answer to Q1**
>
> We construct a working adjacency matrix $A$ online. In the *deterministic* setting, when a new transition $i\ \to j$ is observed we set $A_{ij}=1$. Before option extraction, we row-normalize $A$ to obtain a transition matrix (rows sum to 1). In the *stochastic* setting, to encode the upward “wind,” for actions that move the agent **up** we instead set $A_{ij}=1+\tfrac{1}{4d_i}$, where $d_i$ is the current degree of node $i$. Because $d_i$ can change as the working graph grows, these weights are recomputed prior to each option extraction. Thus, the agent has knowledge of the wind (as in *diffusion options* [10]) but not of the full graph. Policy extraction follows Algorithm 1 (line 10): at each state $s$, the intra-option policy $\pi$ deterministically selects the action whose successor-state representation has the largest magnitude. A stochastic policy variant of NEO performs comparably: given an option eigenvector $f$, we can employ the policy
> $\pi(a \mid s)\=
> \frac{\lvert f\bigl(T(s,a)\bigr)\rvert}
>      {\displaystyle\sum_{a'\in\mathcal A(s)} \lvert f\bigl(T(s,a')\bigr)\rvert},$
> where $T(s,a)$ is the successor state of $s$ under action $a$ and $\mathcal A(s)$ is the set of actions available in $s$.
> over 20 independent runs, the policy visits an average of 9,399 out of 10,000 nodes in the NYC domain, a 93.99 % coverage, while achieving complete (100 %) coverage in the Four-Rooms environment.
>
>
> ---
>
> **Answer to Q2**
>
> Thanks for noting the assumption!, the clarification on the bound is acknowledged.
>
> ---
>
> **Answer to Q3**
>
> We chose to focus on cosine similarity (between the oracle eigenvectors and those learned with a PixelCNN) to verify that they can be learned at high accuracy. This also lets us check whether certain eigenvectors are easier to learn, e.g., are the smoothest easier to learn than those at dimension 10-11? We find it important to run these experiments in the same manner as has been done for the standard Laplacian[2] for the following reasons: 1) to ensure that the loss functions in the Proper Laplacian Representation method [2] apply well in our case. 2) Laplacian representations can provide smooth orthogonal representations which are useful empirically, yet, are not the eigenvectors of the Laplacian [8][9]. We want to ensure that the vectors learnt are closely matching the eigenvectors of our source Laplacian rather than being smooth orthogonal representation with magnitude concentration at source nodes - capturing close approximations to the true eigenvectors we believe is highly important as any further properties found (such as the gradient guarantees) can then apply approximately in Hilbert space. For future work, we aim to capture directed versions of these representations from a similar implementation and repeat the cosine-similarity experiments to confirm we can obtain the needed accuracy before extending to asymmetric domains such as Atari games with reward-based and coverage-based evaluation metrics.
>
> ---
>
> **Answer to Q4**
>
> Thanks, we are glad that the figure resolved the sensitivity question.
>
> ---
>
> **Answer to Q5**
>
> We reported the time for a single option discovery call in seconds, not an entire run or an epoch. The reason is that runtime depends on the size of the working graph (eigenvector computation and path distances scale with graph size), so methods that explore more states would otherwise be penalized. It doesn’t take long, so we can include full-run times as well, but keep in mind the above penalization effect:
>
> Average runtime **over 10 runs** (elapsed time in seconds):
>
> | Method  | $\|V\|$  | elapsed time (s) |
> | ------------- | ----: | ---------------: |
> | NEO           |  5099 |            154.9 |
> | cover options |  2155 |            291.9 |
> | eigen options |  2532 |            121.7 |
> | SP-novelty    |  2067 |            117.4 |
>
> ($\|V\|$ is the final number of nodes in the working graph)
>
> On relative performance details: we expect our method to perform similarly to eigen options, except in directed experiments where eigen options requires a polar decomposition, increasing computational time. cover time options recursively computes the Fiedler vector and perform extra operations. In all of our option implementations, we compute only the smoothest $k$ eigenvectors we need, which improves compute speed.
>
> [2] Gomez et al. Proper Laplacian Representation Learning. ICLR, 2024.
>
> [8] Wu et al. The Laplacian in RL: Learning Representations with Efficient Approximations. arXiv 2018.
>
> [9] Wang et al. Towards Better Laplacian Representation in Reinforcement Learning with Generalized Graph Drawing. arXiv 2021.
>
> [10] Bar et al. Option Discovery in the Absence of Rewards with Manifold Analysis. arXiv 2020.

---

> > ### Comment · Reviewer_jL2w · 2025-08-08
> >
> > I thank the authors for the detailed rebuttal. The additional clarifications and experiments have resolved my main concerns, and I have raised my score accordingly.

---

> > > ### Author Response · Authors · 2025-08-08
> > >
> > > Dear Reviewer jL2w,
> > >
> > > Thank you very much for your thorough review, valuable questions, and your careful attention to the assumptions in Theorem 3.2. We are pleased that your concerns have been resolved and that you have decided to increase the score.
> > >
> > > Best wishes!

---

### Official Review · Reviewer_xptd · 2025-07-03

**Clarity:** 3
**Significance:** 3
**Originality:** 3
**Rating:** 4
**Confidence:** 3

**Summary:**

This paper presents NEO, an exploration method that leverages the orthogonality of the graph Laplacian's smoothest eigenvectors to extract exploration options. Unlike prior methods which assume an undirected graph, NEO is made compatible with strongly connected directed graphs by limiting eigenvectors to those with a real eigenvalue below a certain novelty threshold. Compared to prior option papers, NEO's options explicitly drive the agent to more novel regions. Experiments on various undirected and directed domains demonstrate that NEO outperforms exploration baselines.

**Questions:**

Questions

1. "Count-based exploration methods suffer from lag in exploration and eigenoptions can drive back to frequently visited states" - does this materialize in experiments?
2. How is the novelty-based threshold chosen? Does it vary across environments?
3. Is there a more principled method for choosing the hyperparameters for the raw visitation count scaling function?

**Ethical Concerns:**

["NO or VERY MINOR ethics concerns only"]

**Final Justification:**

I have improved my score to recommend acceptance given the authors' clarifications on their method's improvement over eigenoptions conceptually. The authors have also given sufficient explanation and evidence for NEO's potential to be applied to continuous domains, although they have not shown it directly in the experiments yet. Overall, these initial primary concerns have been alleviated from the rebuttal period.

**Limitations:**

Yes

**Quality:**

2

**Strengths And Weaknesses:**

Strengths

1. The visualizations and toy experiments are compelling case studies that clearly demonstrate that the field of the smoothest eigenvectors corresponds to the more novel regions, illustrating how the method can be used for exploration.
2. The paper provides solid theoretical support for the proposed method. With small insight on the eigenvalues and modification of the graph Laplacian, the method is extendable to directed graphs, making it more practical.



Neutral

1. While the writing overall is clear, it is hard to pinpoint what are the specific contributions of the paper. It would be beneficial to highlight them in the introduction.



Weaknesses

1. There is a lack of analysis for why the baselines perform poorly relative to NEO. For instance, "Eigen" gets very low performance despite being intuitively similar to NEO, as it also extracts options from eigenvectors of the graph Laplacian.
2. The method is not compatible with continuous environments, limiting its usefulness. Furthermore, the requirement of having an approximation of the state transition graph presents a chicken-and-egg problem, where you need good exploration to approximate this graph and you need a good approximation of the graph to perform exploration via NEO. Older works [1] that obtain options from the successor representation for exploration already had some mechanism to bootstrap the approximation of this graph as well as some initial work on extensions to continuous domains.



Formatting errors impacting readability:

216: Citations for Shortest Path Novelty is broken.

219: Missing dash for Cover options.

221: Epoch is capitalized.

226, 235: figure 2 is not capitalized.

255: figure 3 is not capitalized.

272: "show" should be "shown".

283: "cature" should be "captured." Space between "atleast".

Figure 3: the symbols and colors for the methods changes across environments, making it hard to compare.


[1] Tomar et al. Successor Options: An Option Discovery Framework for Reinforcement Learning. IJCAI 2019.

---

> ### Author Rebuttal · Authors · 2025-07-31
>
> We thank the reviewer for reading our paper and taking the time to write thoughtful strengths and weaknesses.
> # Rebuttal
>
> ## Weaknesses
>
> **Weakness 1. There is a lack of analysis for why the baselines perform poorly relative to NEO. For instance, "Eigen" gets very low performance despite being intuitively similar to NEO, as it also extracts options from eigenvectors of the graph Laplacian.**
> Thank you for the feedback! We include the node count in Figure 4 to evaluate the total number of nodes visited by each method. We will include the visitation matrix plots from which the node counts are obtained in the appendix to further showcase that eigenoptions and covertimes get stuck cycling previously visited states. Although eigenoptions are derived from Laplacian eigenvectors, the resulting behavioural policies do not induce reliable gradients toward distinct states, as is the case for our method. Even the simplest non-trivial eigenvector, the Fiedler eigenvector, is known to fail under certain conditions, such as directionality or multiple graph components, and higher modes are harder to interpret. Moreover, eigenoptions are sign-symmetric: for any option that moves toward a novel region, an opposite-sign option can to undo that progress, driving agents back to frequently visited states. As working graphs increase in node size within our experiments, small local changes will have diminishing impact on global eigenvectors used for eigen options, so the induced policies change little, and this contributes to agents under these policies revisiting the same regions. In our evaluations, eigenoptions and cover-time options underperform a simple, intuitive policy to go to the most novel known state. By contrast, our method’s eigenvectors provide guarantee gradients, avoid sign-symmetric option pairs and produce less stagnant policies in large domains as novelty counts are updated; these differences explain NEO’s performance gap over these baselines.
>
> **Weakness 2. The method is not compatible with continuous environments, limiting its usefulness. Furthermore, the requirement of having an approximation of the state transition graph presents a chicken-and-egg problem, where you need good exploration to approximate this graph, and you need a good approximation of the graph to perform exploration via NEO. Older works \[1] that obtain options from the successor representation for exploration already had some mechanism to bootstrap the approximation of this graph, as well as some initial work on extensions to continuous domains.**
> We evaluate in domains where exact eigenvectors are computable, so both baselines (eigenoptions, cover options) and our method are assessed using the eigenvectors they are theoretically intended to have. This does not preclude extensions to continuous domains; however, the distinction between symmetric and directed settings is crucial. As detailed in the appendix, the symmetric case is straightforward: the approach in the paper proper Laplacian representations \[3] learns eigenvectors in Hilbert-space for \$L = I - P\$, while our approach uses \$L\_{\Gamma} = I + \Gamma - P\$, where \$\Gamma\$ encodes sink weights that shrink eigenvector values toward zero in proportion to sink strength. Practically, one can add a mean-squared-error (MSE) sink loss (zero at sink states) to pull eigenvector values toward zero with a weight given by the sink strength.
>
> By contrast, extending to the directed setting is notably more challenging. For instance, \[2] describe the extension of eigenvector-based distances to directed graphs as a “highly non-trivial challenge” and notes that, “due to the complex-valued eigenvalues of directed graph Laplacian matrices, designing an optimisation objective to approximate the eigenvectors would be difficult.” Beyond our directed gradient guarantees, we make progress by showing in the appendix that our evaluations recover real eigenpairs and by providing an eigenvalue bound induced by the sink weights. In future work, we aim to scale to large **directed** continuous settings by using sinks to bound the imaginary parts of eigenvectors, thereby removing the central obstacle of approximating complex-valued quantities. Whilst this a challenging problem, our work makes important steps in this direction where no other approach to our knowledge provides a potential route to an end-to-end optimisation solution for directed eigenvectors.
>
> We agree that the “chicken-and-egg” issue is important: effective exploration needs strong representations, yet those representations typically require exploration. Our approach tackles this by coupling exploration with an eigenvector approach, so the learned signals both drive exploration and increasingly reflect the domain’s geometry. Because orthogonality constraints are particularly powerful for representation learning, eigenvector-based methods are natural candidates to address this circularity. However, as our evaluation shows, existing eigenvector option methods underperform on exploration—especially when contrasted with the more intuitive strategy of moving toward novel states.
>
>
> ## Questions
>
> **Q1. “Count-based exploration methods suffer from lag in exploration and eigenoptions can drive back to frequently visited states” — does this materialize in experiments?**
> **Yes.** In Figure 4 (node-count evaluations), eigenoptions *start to plateau after initial exploration* in *Double-Maze* and *Double-Hex*, indicating limited discovery of new states. Since exploration of new states starts to plateau in evaluations, it must be concluded that the agent under eigen options moves around frequently visited states more over time. Mechanistically, signed option pairs tend to undo progress toward novelty, and as the graph grows, the Laplacian eigenvectors change little, so the induced behaviours keep revisiting the same regions. This is a crucial issue as it means that eigen options tend to cycle between frequently visited states more as the exploration challenge increases.
>
> **Q2. How is the novelty-based threshold chosen? Does it vary across environments?**
> In the main paper we use a single global threshold \$\delta\$ across all environments (i.e., no per-task tuning). In the appendix, we ablate this choice by sweeping \$\delta\$. Conceptually, \$\delta\$ is the *base novelty cutoff* in our scaling. As illustrated in Figure 1, with \$\delta = 0.5\$ any eigenvalue \$\lambda \le \delta\$ (e.g., \$\lambda = 0.4866\$) satisfies our gradient guarantee. When \$\lambda > \delta\$, it exceeds the scaled value assigned to the least-novel state, so the sufficient condition used in our proof no longer holds; specifically, the term \$1/(1 + N(i) - \lambda)\$ is not guaranteed to be \$< 1\$ for any states \$i\$.
>
> **Q3. Is there a more principled method for choosing the hyperparameters for the raw visitation count scaling function?**
> In the appendix, we show robustness across a range of hyperparameters (Figure 3) and evaluate a simple default: \$\delta = 1, k = 1\$. For a more principled choice, in directed/stochastic domains, the imaginary part of eigenvalues is \$\beta\$-bounded; where it is of interest to avoid imaginary components, a higher \$\delta\$ can be chosen. In undirected (symmetric) domains, a range of values provides similar performance.
>
> [3] Wang et al. Reachability-Aware Laplacian Representation in Reinforcement Learning. arXiv 2022.

---

> > ### Comment · Reviewer_xptd · 2025-08-01
> >
> > > Weakness 1.
> >
> > Thank you for providing more detailed intuition on eigenoptions' failure modes. Yes, it would be much clearer to the reader to provide visualizations on these failure modes, such as the cycling of previously visited states, and how NEO overcomes the same cases. As a side note, it would also be helpful to reference and discuss Figure 4 in the text.
> >
> > > Weakness 2.
> >
> > Thanks for the references to the details on extending the method to the continuous domain in different settings. This primarily clears up my concern regarding compatibility with continuous environments. For the symmetric case, is the extension straightforward to execute empirically? Or do you foresee any potential pitfalls / limitations of the proposed approach for symmetric continuous domains?
> >
> > > Question 1.
> >
> > The general reasoning, "moves around frequently visited states more over time", makes sense. However, do you have any analysis to show that the signed option pairs are _directly_ undoing progress by moving back to frequently visited states, versus the other case, where the options are simply noisy, causing the agent to randomly move around and therefore, have low probability of encountering novel states?
> >
> > > Question 3.
> >
> > Thanks for the reference. The figure is convincing regarding the method's robustness to the two hyperparameters for the scaling function.

---

> > > ### Author Response · Authors · 2025-08-08
> > >
> > > We thank the reviewer for their comments and questions.
> > >
> > > **Question 1 (& Weakness 1)**
> > >
> > > Here we present an analysis focusing on the Fiedler vector $f$ that is required for both cover options and eigen options methods in symmetric settings.
> > > Its two extrema define the boundary states: $
> > > b_1=\arg\max f$, and $b_2=\arg\min f$. By the **hot-spot conjecture** [6], these extrema drift toward geometric extremes of the domain. On an $n\times n$ grid, for example, they would occupy opposite corners.
> > >
> > > ---
> > >
> > > ### Why one boundary of the Fiedler vector tends to sit in a well-visited region
> > >
> > > Consider an online agent which begins close to one corner and incrementally builds a graph; that corner area is likely to quickly become the most connected, most frequently visited area.
> > > Placing one boundary (say $b_1$) there tends to be part of the first non-trivial solution to the Rayleigh quotient $
> > > \frac{\sum_{(u,v)}\bigl(f(u)-f(v)\bigr)^{2}}{f^\top f},
> > > $
> > >
> > > because the dense local connectivity allows $f$ to taper off gradually, keeping each edge difference and thus the numerator small. If the second boundary were nearby, neighbouring nodes would carry opposite extreme values, the numerator would spike, and the quotient would rise. Minimising the quotient therefore pushes the second boundary to a geometrically far away state, such as abother corner, rarely visited.
> > >
> > > ---
> > >
> > > **Analysis:**
> > >
> > > After every run of the 250 k-step experiment in the paper (Fig. 4), we freeze the exploration graph, freeze visitation counts N(.), compute eigen-options, and extract the Fiedler vector $f$. Let $b_{high}$ be the boundary with the larger visit count, $b_{high} = max( N(b1), N(b2) )$, and $b_{low} = min( N(b1), N(b2) )$.
> > > To see whether one sign of this pair tends to “undo progress”, we start the agent at the boundary state with the lowest visitation count ($b_{low}$) and launch the Fiedler vector option that has an increasing gradient from that state, which will cause the agent to move from the lower-visitation boundary toward the opposite boundary until its gradient stalls in a local or global maxima. The results are presented in the table below for two domains. The table shows $b_{low}$, $b_{high}$, the boundary gap, and the visitation count N(.) at option termination state. The values shown are means and medians over 20 runs of the experiment.
> > >
> > > | Domain         | $b_{low}$ (option start state) | $b_{high}$                      | Boundary gap ($b_{high}-b_{low}$) | Visitation count (N) at option termination state |
> > > | -------------- | ------------------------------------ | ---------------------------------- | ---------------------------------------- | ----------------------------------------------- |
> > > | **Large Maze** | mean **129.8**, median **39.0**      | mean **1355.1**, median **1070.5** | mean **1225.3**, median **952.5**        | mean **985.6**, median **638.0**                |
> > > | **Hex**        | mean **68.5**, median **14.5**       | mean **958.9**, median **503.5**   | mean **890.5**, median **467.0**         | mean **822.6**, median **745.0**                |
> > >
> > >
> > > In both domains, one of the signed options lifts the agent from relatively novel states to frequently visited states (e.g. mean 68.5 → 822.6 visits, median 14.5 → 745.0 visits in Hex). We know that this particular option (taking the agent back to a frequently visited state) would naturally be selected by the agent with the same probability as other options. This shows that, if the agent did make progress due to taking the Fiedler vector option that took the agent to a low visitation boundary, then initializing the other option there (which probabilistically occurs) takes the agent backwards to a high visitation state, directly undoing progress.
> > >
> > > **Weakness 2**
> > >
> > > Yes, extending our method to symmetric continuous domains is now quite direct. Recent advances in deep pseudo-count estimation \[7] and Laplacian representation learning [2] remove the bottlenecks that existed only a few years ago. As we note in our response to Reviewer 3, a pixel-CNN encoder already reproduces NEO Laplacian eigenvectors with a median cosine similarity of 0.9907 to the oracle, showing that the necessary eigenvectors can be learned entirely end-to-end. In our appendix, we discuss how skewing the replay distribution toward uniform coverage would be useful in making accurate approximations (this is true in general for other methods such as successor representations, eigenoptions, covertime options). We expect utilizing pseudo-count estimation to obtain more uniform sampling would be additionally useful here. However the data skewness is an optimisation detail, not a conceptual or implementation limitation.
> > >
> > > [2] Gomez et al. Proper Laplacian Representation Learning. ICLR, 2024.
> > >
> > > [6] Bañuelos et al. On the “Hot Spots” Conjecture of J. Rauch. Journal of Functional Analysis, 164, 1999.
> > >
> > > [7] Yang, K., Tao, J., Lyu, J., & Li, X. Exploration and Anti-Exploration with Distributional Random Network Distillation. arXiv 2024.

---

> > > > ### Comment · Reviewer_xptd · 2025-08-08
> > > >
> > > > Thank you for the intuition and more detailed analysis regarding the Fiedler vector. It is now more clearly shown the limitations of eigenoptions and more motivation for the proposed method. It would be beneficial to include this in the paper. Regarding symmetric continuous domains, I appreciate the references to more recent advances and experiments with the parameterized encoder. It would be interesting to demonstrate this empirically in future work.
> > > >
> > > > Given that my concerns regarding comparison to eigenoptions and limited application to continuous domains have been alleviated, I will improve my score.

---

> ### Author Response · Authors · 2025-08-08
>
> Dear Reviewer xptd
>
> Thank you for your insightful questions and feedback! We will update the manuscript to include the detail on the limitations of the Fiedler vector and eigenoptions. We are glad that your concerns have been alleviated, and we appreciate your revised score.
>
> All the best!

---

### Official Review · Reviewer_CwLe · 2025-07-08

**Clarity:** 3
**Significance:** 4
**Originality:** 3
**Rating:** 5
**Confidence:** 1

**Summary:**

The paper proposes a technique to improve exploration in reinforcement learning by using global signals from structures like the eigenspectrum of the graph to guide exploration. The key idea is to use the eigenvectors to construct options by following the gradient of the eigenvector to move in the state space. Starting with this intuition the proposed approach takes into account visitation counts to make the options more effective with respect to exploration. Extensive evaluation across multiple domains and other spectral baselines bear out this improvement in exploration performance.

**Questions:**

- See weaknesses above.

**Ethical Concerns:**

["NO or VERY MINOR ethics concerns only"]

**Quality:**

3

**Strengths And Weaknesses:**

+ Conceptually the method is quite elegant and simple
+ Seems to work very well in practice
+ The paper does a good job of spelling out all the implementation nuances in sufficient detail
+ The paper has a strong set of theoretical results showing that the proposed approach will take us towards more novel states

Weaknesses

- It would be interesting to explore how the method might generalize to scenarios beyond tabular RL and high dimensional inputs, and scenarios where the state transition graph might not be known in advance and have to be constructed e.g. in a partially observable setting. Would be interesting if the paper discussed some of these practical issues that practitioners often care about in practice.

---

> ### Author Rebuttal · Authors · 2025-07-31
>
> Thank you very much for your time and your thoughtful comments.
>
> **Questions:**
>
> Our setting does not assume a known graph: the agent adds nodes and edges to the graph through online exploration. During most of the training, the graph is partially observed and often only weakly connected, which motivates a directed formulation of our option method. We discuss the continuous/high-dimensional case in the appendix, noting that our Laplacian formulation is closely aligned with Laplacian representation learning from recent work in high-dimensional Hilbert space [2]. In particular, we discuss how our method differs in construction only by the addition of “sinks” that enforce a zero potential on sink nodes - this addition is straightforward to perform via an MSE penalty for neural network based solutions. We discuss the directed extension in the appendix as well. This extension is omitted from related spectral approaches due to the difficulty of the problem, yet it is crucial in practice: many robotic domains have asymmetric (directed) dynamics. Starting from a directed-theory foundation while staying compatible with Laplacian-style representations makes the approach naturally extensible to high dimensions in the symmetric setting and potentially extendable in the directed setting. In future work, we will work on scaling this to learning directed eigenvectors in high-dimensional directed problems such as robotic domains with strong directed components.
>
> [2] Gomez et al. Proper Laplacian Representation Learning. ICLR, 2024.

---

### Comment · Area_Chair_gXFx · 2025-08-03
**Reviewers please respond to the rebuttal!**

Dear reviewers,

if you have not yet responded to the rebuttal of the authors, please do so as soon as possible, since the rebuttal window closes soon.

Please check whether all your concerns have been addressed!  If yes, please consider raising your score.

Best wishes,
your AC

---

### Decision · Program_Chairs · 2025-09-17

**Decision:**

Accept (poster)

**Comment:**

The paper proposes to use certain Laplacian matrices to guide exploration in discrete tabular reinforcement learning problems, provide the theory that the methods should work in principle and present experiments that the method does indeed work.  The paper is well written.  All reviewers agree that the paper should be accepted (one borderline accept), solid work!